# Boosting Semi-Supervised 2D Human Pose Estimation by Revisiting Data Augmentation and Consistency Training

## Abstract

The 2D human pose estimation (HPE) is a basic visual problem. However, its supervised learning requires massive keypoint labels, which is labor-intensive to collect. Thus, we aim at boosting a pose estimator by excavating extra unlabeled data with semi-supervised learning (SSL). Most previous SSHPE methods are consistency-based and strive to maintain consistent outputs for differently augmented inputs. Under this genre, we find that SSHPE can be boosted from two cores: advanced data augmentations and concise consistency training ways. Specifically, for the first core, we discover the synergistic effects of existing augmentations, and reveal novel paradigms for conveniently producing new superior HPE-oriented augmentations which can more effectively add noise on unlabeled samples. We can therefore establish paired easy-hard augmentations with larger difficulty gaps. For the second core, we propose to repeatedly augment unlabeled images with diverse hard augmentations, and generate multi-path predictions sequentially for optimizing multi-losses in a single network. This simple and compact design is interpretable, and easily benefits from newly found augmentations. Comparing to state-of-the-art SSL approaches, our method brings substantial improvements on public datasets. Code will be released for academic use.

## 1 Introduction

The 2D human pose estimation (HPE) aims to detect and represent human parts as sparse 2D keypoint locations in RGB images. It is the basis of many visual tasks such as action recognition (Yan et al., 2018; Duan et al., 2022), person re-identification (Zhao et al., 2017; Sarfraz et al., 2018), 3D pose lifting (Nie et al., 2023; Dabhi et al., 2024) and 3D human shape regression (Pavlakos et al., 2018; 2019). Modern data-driven HPE has been substantially improved by generous deep supervised learning approaches (Cao et al., 2017; Cheng et al., 2020; Xu et al., 2022; Yang et al., 2023; Tan et al., 2024). This greatly benefits from the collection and annotation of many large-scale public HPE datasets (Andriluka et al., 2014; Lin et al., 2014; Wu et al., 2019). However, compared to image classification and detection tasks requiring plain labels, obtaining accurate 2D keypoints from massive images is laborious and time-consuming. To this end, some researches (Xie et al., 2021; Moskvyak et al., 2021; Wang et al., 2022; Huang et al., 2023; Yu et al., 2024) try to alleviate this problem by introducing the semi-supervised 2D human pose estimation (SSHPE), which can subtly leverage extensive easier obtainable yet unlabeled 2D human images in addition to partial labeled data to improve performance. Although methods (Xie et al., 2021; Huang et al., 2023) have improved the accuracy of SSHPE task, they overlooked two fundamental questions:

Q1: **How to judge the discrepancy of unsupervised data augmentations with different difficulty levels?** As shown in Fig. 1a, for a batch of unlabeled images $\mathbf{I}$, its easy augmentation $\mathbf{I}_e$ and hard augmentation $\mathbf{I}_h$ are generated separately in (Xie et al., 2021). Then, predicted heatmap $\mathbf{H}_e$ of $\mathbf{I}_e$ is used as a pseudo label to teach the network to learn the harder counterpart $\mathbf{I}_h$ with its yielded heatmap $\mathbf{H}_h$. Xie et al. (2021) finds that the large gap between two augmentations $(\mathbf{I}_e, \mathbf{I}_h)$ matters. Essentially, this is a pursuit for more advanced data augmentations in SSHPE. To rank augmentations of different difficulty levels, Xie et al. (2021) observes precision degradation of a pretrained model by testing it on a dataset after corresponding augmentation. However, we declare that this manner is not rigorous. An obvious counter-case is that over-augmented samples approaching noise will get

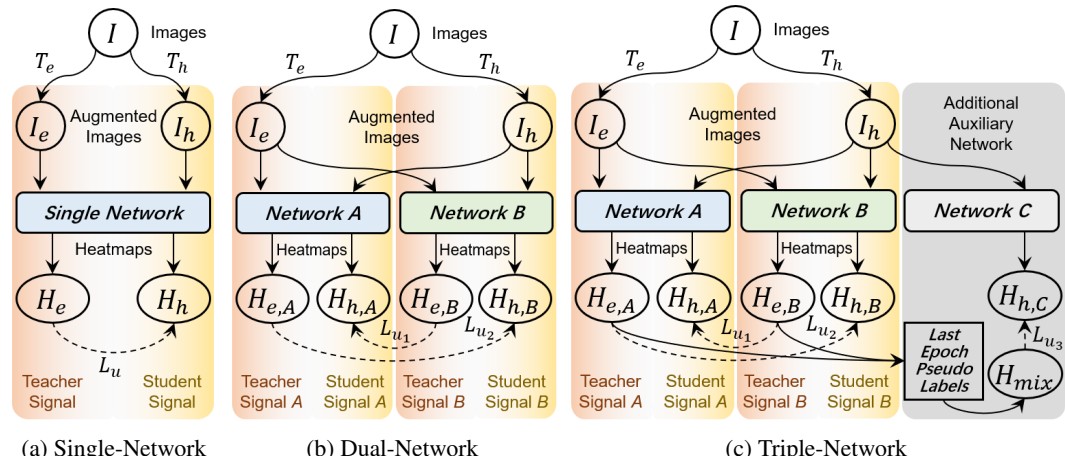

Figure 1: Frameworks of existing semi-supervised human pose estimation (SSHPE) methods including (a) Single-Network and (b) Dual-Network which are proposed by (Xie et al., 2021), and (c) Triple-Network proposed by (Huang et al., 2023).

worst evaluation results, but such hard augmentations are meaningless. Contrastingly, we deem that a persuasive ranking requires independent training for each augmentation. We answer this question in detail in Sec. 3.1.

**Q2**: **How to generate multiple unsupervised signals for consistency training efficiently and concisely?** Previous work (Xie et al., 2021) proposes to use a Single-Network to perform the unsupervised consistency training on the easy-hard pair $(\mathbf{I}_e, \mathbf{I}_h)$. It also gives a more complicated Dual-Network as in Fig. 1b for cross-training of two easy-hard pairs. SSPCM (Huang et al., 2023) even constructs a Triple-Network as in Fig. 1c for interactive training of three easy-hard pairs. This pattern of adding more networks with the increase of unsupervised signal pairs can certainly bring gains. But this is cumbersome and will decelerate the training speed proportionally. In practice, considering that augmentations are always performed on the same input, we can repeatedly augment $\mathbf{I}$ multiple times with $n$ diverse hard augmentations, and generate multi-path easy-hard pairs $\{(\mathbf{I}_e, \mathbf{I}_{h_1}), (\mathbf{I}_e, \mathbf{I}_{h_2}), ..., (\mathbf{I}_e, \mathbf{I}_{h_n})\}$. In this way, we can use only one single network (refer Fig. 6a) to optimize $n$ pairs of losses. This is also applicable to dual networks (refer Fig. 6b). We discuss this question in detail in Sec. 3.2.

In this paper, we mainly revisit Q1 and Q2 to boost SSHPE. For Q1, after properly ranking existing basic augmentations (DeVries & Taylor, 2017; Zhang et al., 2018; Yun et al., 2019; Cubuk et al., 2018; 2020), we naturally try to extend it to discover new strong augmentations through reasonable sequential combinations inspired by the AutoAug families (Cubuk et al., 2018; Lim et al., 2019; Hataya et al., 2020; Zheng et al., 2022). Rather than trivial enumeration, we notice the **synergistic effects** between different augmentations, and reveal novel paradigms for easily generating superior combinations. Our paradigms for the SSHPE task contain three principles: (P1) *Do not combine MixUp-related augmentations.* (P2) *Try to utilize the synergistic effects.* (P3) *Do not over-combine too many augmentations.* These principles have favorable interpretability, and bypass painstaking designs in other advanced augmentations (Hendrycks et al., 2020; Müller & Hutter, 2021; Zheng et al., 2022; Pinto et al., 2022; Liu et al., 2022; Han et al., 2022). For Q2, we quantitatively validated the superiority of multi-path design over commonly used heatmaps fusion (Radosavovic et al., 2018) and confidence masking Xie et al. (2020a); Huang et al. (2023). Combining it with newly found advanced augmentations, our Single-Network based approach can surpass the original Dual-Network (Xie et al., 2021) and come close to SSPCM using a Triple-Network.

In summary, our contributions are three-folds: (1) We comprehensively evaluated the difficulty levels of existing data augmentations suitable for the SSHPE task, validated their synergistic effects by properly combining different basic augmentations, and presented novel combination paradigms which are intuitively interpretable. (2) We proposed to generate multi-path predictions of separately strongly augmented samples for training only one single model, rather than adding auxiliary networks. Thus, we can optimize multiple unsupervised losses efficiently and concisely, and benefit

from distinct superior augmentations. (3) We achieved new SOTA results on public SSHPE benchmarks with less training time and parameters under same settings of previous methods.

## 2 RELATED WORK

**Semi-Supervised Learning (SSL)** originated in the classification task by exploiting a small set of labeled data and a large set of unlabeled data. It can be categorized into pseudo-label (PL) based (Radosavovic et al., 2018; Oliver et al., 2018; Xie et al., 2020b; Sohn et al., 2020; Guo & Li, 2022; Wang et al., 2023) and consistency-based (Laine & Aila, 2016; Tarvainen & Valpola, 2017; Berthelot et al., 2019; Xie et al., 2020a; Zhang et al., 2021; Gui et al., 2023; Huang et al., 2024). PL-based methods iteratively add unlabeled images into the training data by pseudo-labeling them with a pretrained or gradually enhanced model. It needs to find suitable thresholds to mask out uncertain samples with low-confidence, which is a crucial yet tricky issue. Consistency-based methods enforce model outputs to be consistent when its input is randomly perturbed. They have shown to work well on many tasks. For example, MixMatch (Berthelot et al., 2019) combines the consistency regularization with the entropy minimization to obtain confident predictions. FixMatch (Sohn et al., 2020) utilizes a weak-to-strong consistency regularization and integrates the pseudo-labeling to leverage unlabeled data. FlexMatch (Zhang et al., 2021) and FreeMatch (Wang et al., 2023) adopt the curriculum learning and adaptive thresholding based on FixMatch, respectively. CRMatch (Fan et al., 2023) and SAA (Gui et al., 2023) try to design strategies and augmentations to enhance the consistency training. These SSL methods give us primitive inspirations.

**Semi-Supervised Human Pose Estimation (SSHPE)** is relatively less-studied comparing to other visual tasks classification and object detection. A few SSHPE methods are based on pseudo labeling (Wang et al., 2022; Springstein et al., 2022) or consistency training (Xie et al., 2021; Moskvyak et al., 2021; Li & Lee, 2023; Huang et al., 2023). SSKL (Moskvyak et al., 2021) designs a semantic keypoint consistency constraint to learn invariant representations of same keypoints. It has been evaluated on small-scale HPE benchmarks MPII (Andriluka et al., 2014) and LSP (Johnson & Everingham, 2011), instead of the larger COCO (Lin et al., 2014). Following it, PLACL (Wang et al., 2022) introduces the curriculum learning by auto-selecting dynamic thresholds for producing pseudo-labels via reinforcement learning. Inspired by co-training (Qiao et al., 2018) and dual-student (Ke et al., 2019), Dual-Network (Xie et al., 2021) points out the typical collapsing problem in SSHPE, and proposes the easy-hard augmentation pair on the same input to imitate teacher-student signals without relying on Mean Teacher (Tarvainen & Valpola, 2017). SSPCM (Huang et al., 2023) extends the Dual into Triple by adding an auxiliary teacher for interactive training in multi-steps. It designs a handcrafted pseudo-label correction based on the predicted position inconsistency of two teachers, and has achieved SOTA performances. Still based on Dual-Network, Pesudo-HMs (Yu et al., 2024) utilizes the cross-student uncertainty to propose a threshold-and-refine procedure, which can denoise and select reliable pseudo-heatmaps as targets for learning from unlabeled data. While, in this paper, we revisit the less efficient consistency training way in (Xie et al., 2021; Huang et al., 2023), and propose to upgrade the Single-Network by multi-path predictions.

**Unsupervised Data Augmentations** The UDA (Xie et al., 2020a) has emphasized and verified the key role of high-quality noise injection (*e.g.*, data augmentations) in improving unsupervised consistency training. It utilizes advanced augmentations (Cubuk et al., 2018; 2020) to promote the SSL classification. Then, Xie et al. (2021) transfers the positive correlation between strong augmentations and SSL performance to the HPE field. It introduces a more advanced augmentation called Joint Cutout inspired by Cutout (DeVries & Taylor, 2017). Similarly, SSPCM (Huang et al., 2023) provides a harder keypoints-sensitive augmentation Cut-Occlude inherited from CutMix (Yun et al., 2019). In this paper, we thoroughly revisit existing data augmentations suitable to SSHPE, give a rank of their difficulty levels by controlled trainings, and produce simple paradigms for getting novel superior joint-related augmentations. We also compare them with other well-designed counterparts (Cubuk et al., 2020; Müller & Hutter, 2021; Han et al., 2022) to reveal our advantages.

## 3 EMPIRICAL STUDIES

**Problem Definition**: The task of 2D HPE is to detect $k$ body joints in an image $\mathbf{I} \in \mathbb{R}^{h \times w \times 3}$. The state-of-the-art methods (Xiao et al., 2018; Sun et al., 2019) tend to estimate $k$ Gaussian heatmaps

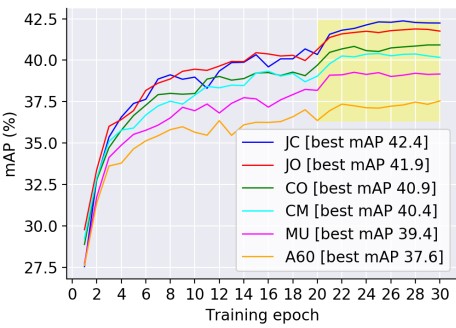
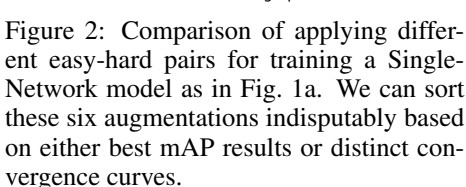
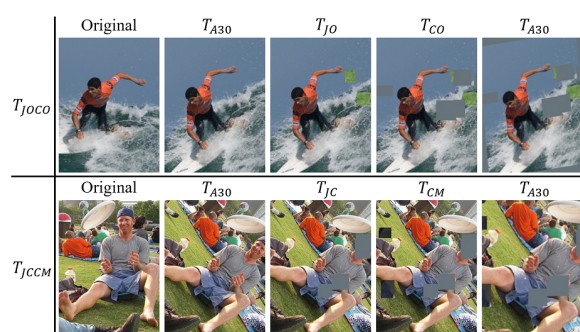

Figure 2: Comparison of applying different easy-hard pairs for training a Single-Network model as in Fig. 1a. We can sort these six augmentations indisputably based on either best mAP results or distinct convergence curves.

Figure 3: Illustrations of two novel superior combinations $T_{JOCO}$ and $T_{JCCM}$. Either of them is a sequential operations using ready-made collaborative augmentation. And $T_{JO}$ and $T_{CM}$ introduce extra patches cropped from other images which are not displayed.

$\mathbf{H} \in \mathbb{R}^{\frac{h}{s} \times \frac{w}{s} \times k}$ downsampled $s$ times. For inference, each keypoint is located by finding the pixel with largest value in its predicted heatmap. We denote the labeled and unlabeled training sets as $\mathcal{D}^l = \{(\mathbf{I}_i^l, \mathbf{H}_i^l)\}|_{i=1}^N$ and $\mathcal{D}^u = \{\mathbf{I}_i^u\}|_{i=1}^M$, respectively. Here, the $\mathbf{I}_i^l$ or $\mathbf{I}_i^u$ means a labeled or unlabeled image sample, respectively. And $N$ or $M$ is the total number of image samples. The $\mathbf{H}^l$ are ground-truth heatmaps generated using 2D keypoints. For supervised training of the network $f$, we calculate the MSE loss:

$$\mathcal{L}_s = \mathbb{E}_{\mathbf{I} \in \mathcal{D}^l} ||f(T_e(\mathbf{I})) - T_e(\mathbf{H})||^2, \tag{1}$$

where $T_e$ represents an easy affine augmentation including a random rotation angle from $[-30°, 30°]$ and scale factor from $[0.75, 1.25]$ (denoted as $T_{A30}$). For unlabeled images, we calculate the unsupervised consistency loss:

$$\mathcal{L}_u = \mathbb{E}_{\mathbf{I} \in \mathcal{D}^u} ||T_{e \to h}(f(T_e(\mathbf{I}))) - f(T_h(\mathbf{I}))||^2, \tag{2}$$

where $T_h$ is a harder augmentation with strong perturbations than affine-based $T_e$. The $T_{e \to h}$ means a known affine transformation on heatmaps if $T_h$ contains additional rotation and scaling operations. In this way, we can obtain a paired *easy-hard* augmentations $(\mathbf{I}_e, \mathbf{I}_h) = (T_e(\mathbf{I}), T_h(\mathbf{I}))$ for generating corresponding teacher signals and student signals. During training, we should stop gradients propagation of teacher signals to avoid collapsing. Next, we answer two questions Q1 and Q2 by extensive empirical studies in Sec. 3.1 and Sec. 3.2, respectively. After that, we provide a theoretical perspective for understanding the pursuit of designing stronger augmentations in Sec. A.1.

### 3.1 PARADIGMS OF GENERATING SUPERIOR AUGMENTATIONS

**Ranking of Basic Augmentations** The core of the easy-hard pair paradigm $(\mathbf{I}_e, \mathbf{I}_h)$ is a more advanced augmentation. For this reason, Dual-Network (Xie et al., 2021) and SSPCM (Huang et al., 2023) propose pseudo keypoint-based augmentations Joint Cutout ($T_{JC}$) and Joint Cut-Occlude ($T_{JO}$), respectively. They also reach a similar yet crude conclusion about difficulty levels of existing augmentations: $\{T_{JO}, T_{JC}\} > \{T_{RA}, T_{CM}, T_{CO}, T_{MU}, T_{A60}\}$, where $T_{RA}$, $T_{CM}$, $T_{CO}$ and $T_{MU}$ are RandAugment (Cubuk et al., 2020), CutMix (Yun et al., 2019), Cutout (DeVries & Taylor, 2017) and Mixup (Zhang et al., 2018), respectively. The $T_{A60}$ consists of two $T_{A30}$. We give them a new ranking by conducting more rigorous trainings one-by-one. The $T_{RA}$ is removed for it contains repetitions with $T_{CO}$ and $T_{A60}$. As shown in Fig. 2, we divide the rest by their distinguishable gaps into four levels: $\{T_{JC}, T_{JO}\} > \{T_{CO}, T_{CM}\} > \{T_{MU}\} > \{T_{A60}\}$.

**Synergy between Augmentations** Then, instead of laboriously designing stronger augmentations, we consider to conduct two or more augmentations in sequence to obtain superior combinations conveniently. This idea is feasible because it essentially belongs to the AutoAug families (Cubuk et al., 2018; Lim et al., 2019; Hataya et al., 2020; Zheng et al., 2022). Instead of auto-searching, we expect to find some heuristic strategies for the HPE task. In fact, after performing joint-related $T_{JO}$ or $T_{JC}$ on one image, we can continue to perform some joint-unrelated augmentations such as $T_{CM}$, $T_{CO}$ and $T_{MU}$ on random areas. As shown in Fig. 3, applying $T_{JOCO}$ (a $T_{CO}$ after $T_{JO}$) or

$T_{JCCM}$ (a $T_{CM}$ after $T_{JC}$) will bring harder samples for generating more effective student signals, but not destroy the semantic information visually. We call this discovery the **synergistic effect** between different augmentations. The $T_{A60}$ can server as an essential factor in any $T_h$ for keeping the geometric diversity.

**Selection of Augmentations Combination** Now, if selecting from the rest five basic augmentations, there are up to 26 choices ($2^5 - \binom{5}{0} - \binom{5}{1}$). The optimal combinations are still time-consuming to acquire. Fortunately, not arbitrary number or kind of augmentations are collaborative. We intuitively summarize three simplistic principles. (P1) A global $T_{MU}$ does not make sense for the HPE task. (P2) Stacking augmentations with the similar perturbation type (e.g., $T_{JO} \sim T_{CM}$ and $T_{JC} \sim T_{CO}$) or difficulty level (e.g., $T_{JO} \sim T_{JC}$ and $T_{CM} \sim T_{CO}$) may not bring significant gain. (P3) Adding too many augmentations (e.g., three or four) will be profitless or even harmful for seriously polluting the image. We thus nominate the most likely superior combinations: $T_{JOCO}$ and $T_{JCCM}$. A case of setting $T_e$ as $T_{A30}$ and $T_h$ as $T_{JOCO}$ for getting corresponding easy teacher signals $\mathbf{H}_e$ and hard student signals $\mathbf{H}_h$ is shown as below:

$$\begin{aligned}
\mathbf{H}_e &= T_{A30 \to A60}(f(\mathbf{I}_e)), \quad \mathbf{I}_e = T_{A30}(\mathbf{I}), \\
\mathbf{H}_h &= f(\mathbf{I}_h), \quad \mathbf{I}_h = T_{A30}(T_{CO}(T_{JO}(T_{A30}(\mathbf{I}))))
\end{aligned} \tag{3}$$

where $T_{A60}$ is the default of $T_h$, and divided into two separate $T_{A30}$ for being compatible with $T_e$. To further verify the above intuitive principles, we follow the augmentations ranking way and conduct empirical studies on the performance of up to 13 selected representative combinations out of 26 choices.

Table 1: Best mAPs of different combinations.

| Id | Combination | mAP |
|----|-------------|-----|
| – | $T_{MU}$ | 39.4 |
| – | $T_{CO}$ | 40.9 |
| – | $T_{CM}$ | 40.4 |
| – | $T_{JC}$ | 42.4 |
| – | $T_{JO}$ | 41.9 |
| $c_1$ | $T_{JC,CM}$ | 42.7 |
| $c_2$ | $T_{JO,CO}$ | **43.7** |
| $c_3$ | $T_{JC,MU}$ | 41.8 |
| $c_4$ | $T_{JO,MU}$ | 42.1 |
| $c_5$ | $T_{JC,CO}$ | 42.1 |
| $c_6$ | $T_{JO,CM}$ | 42.5 |
| $c_7$ | $T_{JC,JO}$ | 42.0 |
| $c_8$ | $T_{JC,CM,MU}$ | 42.0 |
| $c_9$ | $T_{JO,CO,MU}$ | 42.8 |
| $c_{10}$ | $T_{JC,JO,CO}$ | 41.7 |
| $c_{11}$ | $T_{JC,JO,CM}$ | 42.3 |
| $c_{12}$ | $T_{JC,CO,CM}$ | 42.8 |
| $c_{13}$ | $T_{JO,CO,CM}$ | 42.8 |

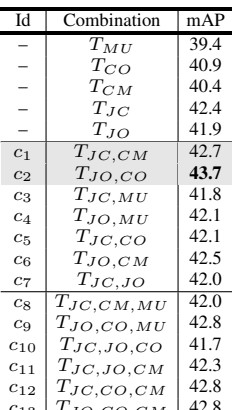

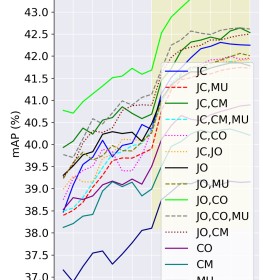

Figure 4: The convergence curves of different tests in Tab. 1.

Table 2: Best mAP results of different MultiAugs tests.

| Id | MultiAugs | mAP |
|----|-----------|-----|
| $m_1$ | $T_{JC}, T_{CM}$ | 43.0 |
| $m_2$ | $T_{JO}, T_{CO}$ | 43.4 |
| $m_3$ | $T_{JC}, T_{JO}$ | 43.1 |
| $m_4$ | $T_{JCCM}, T_{CO}$ | 43.7 |
| $m_5$ | $T_{JOCO}, T_{CM}$ | 43.1 |
| $m_6$ | $T_{JOCO}, T_{JC}$ | 44.2 |
| $m_7$ | $T_{JCCM}, T_{JO}$ | 42.9 |
| $m_8$ | $T_{JC}, T_{JO}, T_{CO}, T_{CM}$ | 43.6 |
| $m_9$ | $T_{JCCM}$ (twice) | 43.6 |
| $m_{10}$ | $T_{JOCO}$ (twice) | 44.0 |
| $m_{11}$ | $T_{JOCO}, T_{JCCM}$ | 44.9 |
| $m_{12}$ | $T_{JOCO}, T_{JC}, T_{JCCM}, T_{JO}$ | **45.5** |

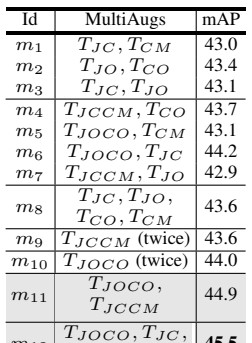

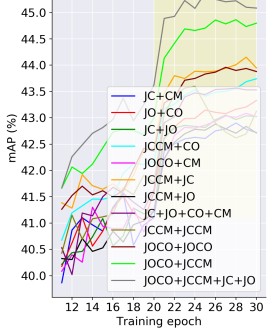

Figure 5: The convergence curves of different tests in Tab. 2.

As shown in Tab. 1 and Fig. 4, we can examine three principles one-by-one: (P1) $T_{MU}$ *often causes adverse or inferior effects for each combination.* Please refer paired combinations $c_1$-$c_3$, $c_2$-$c_4$, $c_1$-$c_8$ and $c_2$-$c_9$. Thus, we do not add it for finding superior combinations. (P2) *Synergistic effects between augmentations do exist.* Please refer paired combinations $c_1$-$c_5$, $c_1$-$c_7$, $c_2$-$c_6$ and $c_2$-$c_7$. Especially, the $T_{JCJO}$ with two most advanced augmentations performs the worst among combinations $\{c_1, c_2, c_5, c_6, c_7\}$, which roundly reveals the harm of violating the principle of synergy. (P3) *Do not overly combine too many augmentations.* Please refer paired combinations $c_1$-$c_{12}$, $c_2$-$c_{13}$, $c_1$-$c_{11}$ and $c_2$-$c_{10}$. Stacking more augmentations brings non-significant gains or even results in degradation. We attribute it to deviating from the rule of collaboration and possibly producing meaningless or difficult-to-recognize images. Based on these facts, we have sufficient reasons not to check the performance of rest combinations, and recommend two new strongest combinations $T_{JOCO}$ and $T_{JCCM}$.

## 3.2 Multi-path Consistency Losses

**Intuitive Motivation** To further amplify the advantage of easy-hard augmentation, Xie et al. (2021) adopts two independent networks containing two easy-hard pairs for producing two consistency losses. SSPCM (Huang et al., 2023) continues this route by designing a Triple-Network with three easy-hard pairs. Meanwhile, some SSL methods construct multi-view inputs for unlabeled data

without adding accompanying networks. For example, SwAV (Caron et al., 2020) enforces the local-to-global consistency among a bag of views with different resolutions. ReMixMatch (Berthelot et al., 2020) feeds multiple strongly augmented versions of an input into the model for training. Therefore, we wonder whether such a simple idea can also benefit the SSHPE task.

Specifically, rather than feeding a single hard augmentation $\mathbf{I}_h$ into the model, we independently yield $n$ strongly augmented inputs $\mathcal{I}^n = \{\mathbf{I}_{h_1}, \mathbf{I}_{h_2}, ..., \mathbf{I}_{h_n}\}$ from $\mathbf{I}$ by applying $n$ hard data augmentations $\mathcal{T}^n = \{T_{h_1}, T_{h_2}, ..., T_{h_n}\}$ accordingly. The augmentation set $\mathcal{T}^n$ is de-emphasized in order and non-deterministic, and will generate distinct multi-path augmented inputs $\mathcal{I}^n$. Then, we can calculate $n$-stream heatmaps $\mathcal{H}^n = \{f(T_{h_i}(\mathbf{I}_{h_i}))|_{i=1}^n\}$. This multi-path augmentation framework is illustrated in Fig. 6. For regularizing $n$ easy-hard pairs, we obtain multi-path consistency losses using Eq. 2 in separate, and optimize them jointly by applying multi-loss learning:

$$\mathcal{L}_u^* = \mathbb{E}_{\mathbf{H}_{h_i} \in \mathcal{H}^n} \sum\nolimits_{i=1}^n ||\mathbf{H}_e - \mathbf{H}_{h_i}||^2, \tag{4}$$

where $\mathbf{H}_e$ and $\mathbf{H}_{h_i}$ are obtained as dissected in Eq. 3. The $\mathbf{H}_e$ keeps constant for each $\mathbf{H}_{h_i}$. Comparably, Data Distill (Radosavovic et al., 2018) applies a single model to multiple transformations of unlabeled data to train a student model. Then, it ensembles predictions to obtain keypoint locations, and re-generates a pseudo heatmap for supervision. Differently, we argue that conducting a fusion on predicted heatmaps in SSHPE is harmful. We consider that there are always differences in the estimation of keypoint positions for each $\mathbf{I}_{h_i}$. It is an ill-posed problem to heuristically evaluate the consistency regularization contribution of each heatmap in pixel during ensemble. We will explain this in ablation studies Sec. 5.3.

Despite the simplicity, such a minor modification brings consistent gains over the original Single-Network under same SSL settings. With our discovered augmentation combinations $T_{JOCO}$ and $T_{JCCM}$, the boosted Single-Network can surpass the original Dual-Network evidently. We validated in ablation studies that the performance gain is non-trivial. We conjecture that regularizing multiple hard augmentations with a shared easy augmentation can be regarded as enforcing consistency among advanced augmentations as well, which inherits the concept of training positive-negative paired samples in contrastive learning (Chen et al., 2020; He et al., 2020; Chen & He, 2021) and its SSL-related variations (Li et al., 2021a; Yang et al., 2022; Wu et al., 2023).

**Effectiveness Verification** As shown in Tab. 2 and Fig. 5, we also experimentally verified the major advantage of the multi-path augmentations (dubbed as MultiAugs) strategy. Here, we have two variables of MultiAugs: the number of paths and the category of augmentations. For acquisition of an optimal augmentations set, similarly, we continue with the rules summarized in the previous section, and elect 12 different MultiAugs schemes for illustrating. We can witness the effectiveness of MultiAugs from two aspects: (1) *It can inherit and even expand the synergistic effects between different augmentations.* (2) *It can alleviate the defects caused by excessive stacking of augmentations.*

Specifically, by comparing $c_1$-$m_1$, $c_2$-$m_2$ and $c_7$-$m_3$, we find that MultiAugs provides comparable performance with the same augmentations. By comparing $c_{10}$-$m_6$, $c_{11}$-$m_7$, $c_{12}$-$m_4$ and $c_{13}$-$m_5$, we can observe the sustained large gains brought by MultiAugs. The scheme $m_8$ does not obtain a better result than $m_{11}$ showing that new augmentations $T_{JOCO}$ and $T_{JCCM}$ have their essential properties. Schemes $m_9$ and $m_{10}$ which utilize a single augmentation twice are also inferior to $m_{11}$ showing the cooperativity of using multi-path distinct augmentations. Moreover, the optimal scheme $m_{12}$ further unleashes capabilities of four advanced augmentations. In order to balance performance and time consumption, we do not add more augmentation paths.

## 4 OUR OVERALL FRAMEWORK: MULTIAUGS

We leverage unlabeled images by applying multiple augmentations with integrating two key techniques introduced in Sec. 3.1 and Sec. 3.2. Firstly, assuming that we have obtained an optimal augmentation set $\widehat{\mathcal{T}}^n = \{\widehat{T}_{h_i}|_{i=1}^n\}$, where $\widehat{T}_{h_i}$ may be an old single augmentation or a novel combined one. Then, we present how to construct our overall training framework in Fig. 6 based on either the Single-Network or the Dual-Network.

**MultiAugs (Single-Network)** It is a consistency-based approach. We only need to maintain a single model as in Fig. 6a during training. For each input batch with equal number of labeled images $\mathbf{I}^l$ and unlabeled images $\mathbf{I}^u$, we calculate the supervised loss with the ground-truth heatmaps as in Eq. 1,

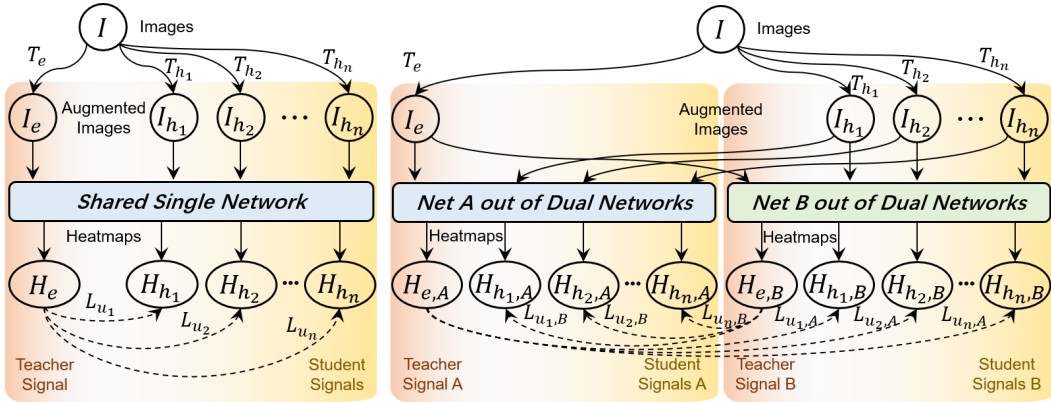

(a) MultiAugs with Single-Network        (b) MultiAugs with Dual-Network

Figure 6: Our proposed MultiAugs uses the (a) Single-Network or (b) Dual-Network, which can utilize multiple hard augmentations and also facilitate multi-path consistency training.

and the multiple unsupervised losses as in Eq. 4, respectively. The final loss is obtained by adding the two loss functions $\mathcal{L} = \mathcal{L}_s + \lambda \mathcal{L}_u^*$ with $\lambda = 1$. Note that we only pass the gradient back through $n$ hard augmentations $\widehat{\mathcal{I}}^n$ for generating teacher signals to avoid collapsing. Based on this boosted Single-Network, we complete all ablation experiments by changing the augmentation categories and quantities in $\widehat{\mathcal{T}}^n$ for controlling the unsupervised loss factor $\mathcal{L}_u^*$.

**MultiAugs (Dual-Network)** As shown in Fig. 6b, this framework learns two identical yet independent networks with each similar to the Single-Network. For one input batch in every step, each of the two networks serves as both a teacher and a student. They both are fed by easy and hard augmentations of unlabeled images $\mathbf{I}^u$ when they produce teacher signals and student signals, respectively. Assuming we have two networks $f_A$ and $f_B$, and also the augmented easy images $\mathbf{I}_e^u$ using $T_e$ and $n$-path hard images $\{\mathbf{I}_{h_1}^u, \mathbf{I}_{h_2}^u, ..., \mathbf{I}_{h_n}^u\}$ using $\widehat{\mathcal{T}}^n$, we first predict the following four types of heatmaps:

$$\mathbf{H}_{e,A} = T_{A30 \to A60}(f_A(\mathbf{I}_e^u)), \mathcal{H}_A = \{\mathbf{H}_{h_i,A}|_{i=1}^n, \mathbf{H}_{h_i,A} = f_A(\mathbf{I}_{h_i}^u)\},$$
$$\mathbf{H}_{e,B} = T_{A30 \to A60}(f_B(\mathbf{I}_e^u)), \mathcal{H}_B = \{\mathbf{H}_{h_i,B}|_{i=1}^n, \mathbf{H}_{h_i,B} = f_B(\mathbf{I}_{h_i}^u)\}, \quad (5)$$

where $T_{A30 \to A60}$ is a pre-generated affine transformation. Based on above heatmaps, we calculate two unsupervised losses for training two networks as follows:

$$\mathcal{L}_{u,A}^* = \mathbb{E}_{\mathbf{H}_{h_i,A} \in \mathcal{H}_A} \sum\nolimits_{i=1}^n ||\mathbf{H}_{e,B} - \mathbf{H}_{h_i,A}||^2,$$
$$\mathcal{L}_{u,B}^* = \mathbb{E}_{\mathbf{H}_{h_i,B} \in \mathcal{H}_B} \sum\nolimits_{i=1}^n ||\mathbf{H}_{e,A} - \mathbf{H}_{h_i,B}||^2, \quad (6)$$

where we swap positions of teacher signals $\mathbf{H}_{e,A}$ and $\mathbf{H}_{e,B}$ for realizing the cross training of networks $f_B$ and $f_A$. Following (Xie et al., 2021), we report the average accuracy of the final two well-trained and performance-approached models. Besides, $f_A$ and $f_B$ can have different structures as in (Xie et al., 2021; Huang et al., 2023), where the large one often helps to distill a better small model, but not vice versa. We do not intend to explore this consensus anymore in this paper.

## 5 EXPERIMENTS

### 5.1 DATASETS AND SETUPS

**COCO** The dataset COCO (Lin et al., 2014) has 4 subsets: *train-set* (118K images), *val-set* (5K images), *test-dev* and *test-challenge*. It is a popular large-scale benchmark for human pose estimation, which contains over 150K annotated people. In addition, there are 123K wild unlabeled images (*wild-set*). We selected the first 1K, 5K and 10K samples from *train-set* as the labeled set. In some experiments, unlabeled data came from the remaining images of *train-set*. In other experiments, we

used the whole *train-set* as the labeled dataset and *wild-set* as the unlabeled dataset. The metric of mAP (Average AP over 10 OKS thresholds) is reported.

**MPII and AI-Challenger** The dataset MPII (Andriluka et al., 2014) has 25K images and 40K person instances with 16 keypoints. The dataset AI-Challenger (AIC) (Wu et al., 2019) *train-set* has 210K images and 370K person instances with 14 keypoints. We use MPII as the labeled set, AIC as the unlabeled set. The metric of PCKh0.5 is reported.

**Implementation Details** For a fair comparison with prior works, we use SimpleBaseline (Xiao et al., 2018) to estimate heatmaps and ResNet (He et al., 2016) and HRNet (Sun et al., 2019) as backbones. The input image size is set to $256 \times 192$. We adopt the PyTorch 1.30 and 4 A100 GPUs with each batch size as 32 for training. The initial learning rate is 1e-3. When training on COCO with 10K labeled data, it decreases to 1e-4 and 1e-5 at epochs 70 and 90, respectively, with a total of 100 epochs. When using 1K or 5K labeled data, total epochs are reduced to 30 or 70, respectively. When training on the complete COCO or MPII+AIC, it drops to 1e-4 and 1e-5 at epochs 220 and 260, respectively, with a total of 300 epochs. When testing, we do not flip horizontally.

For data augmentation settings, we keep the easy augmentation $T_e$ as $T_{A30}$ in all experiments. In Section 3.1 and Section 3.2, we have presented details of finding two novel superior hard augmentations (*e.g.*, $T_{JOCO}$ and $T_{JCCM}$) and recommending the optimal multi-path augmentation set $\widehat{\mathcal{T}}^4$ (please see the scheme $m_{12}$ in Tab. 2), repsectively. Therefore, the hard augmentations set $\widehat{\mathcal{T}}^4 = \{T_{JOCO}, T_{JCCM}, T_{JC}, T_{JO}\}$ is chosen for settings **S1**, **S2** and **S3**. While, the less optimal augmentations set $\widehat{\mathcal{T}}^2 = \{T_{JOCO}, T_{JCCM}\}$ is chosen for settings **S4** and **S5** to balance the performance and training time.

## 5.2 PERFORMANCE COMPARISON

Table 3: AP of different methods on COCO *val-set* when different numbers of labels are used. The backbone of all methods is ResNet18.

| Method | Net. Num. | 1K | 5K | 10K | All |
|---|---|---|---|---|---|
| Supervised (Xiao et al., 2018) | 1 | 31.5 | 46.4 | 51.1 | 67.1 |
| PseudoPose (Xie et al., 2021) | 2 | 37.2 | 50.9 | 56.0 | — |
| DataDistill (Radosavovic et al., 2018) | 2 | 37.6 | 51.6 | 56.6 | — |
| PoseCons (Xie et al., 2021) | 1 | 42.1 | 52.3 | 57.3 | — |
| PoseDual (Xie et al., 2021) | 2 | 44.6 | 55.6 | 59.6 | — |
| SSPCM (Huang et al., 2023) | 3 | 46.9 | 57.5 | 60.7 | — |
| Pseudo-HMs (Yu et al., 2024) | 2 | 47.6 | — | — | — |
| Ours (Single) | 1 | 45.5 | 56.2 | 59.9 | — |
| Ours (Dual) | 2 | **49.7** | **58.8** | **61.8** | — |

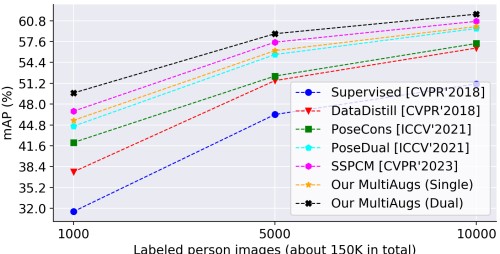

Figure 7: Comparison between state-of-the-art SSHPE methods and our proposed MultiAugs on the COCO dataset.

We mainly compare our MultiAugs with representative SSHPE methods including PoseDual (Xie et al., 2021), SSPCM (Huang et al., 2023) and Pseudo-HMs (Yu et al., 2024) under various conditions. Note that Pseudo-HMs does not follow the same setup as PoseDual (Xie et al., 2021) and SSPCM (Huang et al., 2023), and its code has not been released. We have tried our best to list partial comparable data in some tables to maintain completeness.

**S1**: Firstly, we conduct experiments on the COCO *train-set* with 1K, 5K and 10K labeled data, and evaluate on the *val-set*. As shown in Tab. 3 and Fig. 7, our method brings substantial improvements under the same setting. For example, when using a Single-Network, our method exceeds both PoseCons and PoseDual significantly, and is close to the SSPCM based on three networks. When using a Dual-Network, our method exceeds previous SOTA results by **2.8** mAP, **1.3** mAP, and **1.1** mAP under 1K, 5K and 10K settings, respectively. Note that our method can bring greater gains with less labeled data (*e.g.*, 1K images), which further explains its efficiency and superiority.

**S2**: Then, we conduct larger scale SSHPE experiments on the complete COCO dataset by using *train-set* as the labeled dataset and *wild-set* as the unlabeled dataset. As shown in Tab. 4, regardless of using any backbone, our method can always improve all supervised baseline results, and bring more gains than two compared SSHPE methods (Xie et al., 2021) and (Huang et al., 2023) with using dual networks. When using a single network, our method is still superior to PoseDual (Xie et al., 2021), and fairly close to SSPCM (Huang et al., 2023) based on triple networks.

**S3**: We also report results using HRNet-w48 on the COCO *test-dev* in Tab. 5. The upper, middle and lower panels show CNN-based supervised, Transformers-based supervised, and SSHPE methods, respectively. For SSHPE, COCO *train-set* and *wild-set* is the labeled set and unlabeled set, respectively. Our method can slightly outperform the PoseDual but fall behind the best SSPCM. We attribute it to our fewer training epochs (300 vs. 400) and less parameters (2 Nets vs. 3 Nets) which may lead to weaker generalization. Besides, we can observe that MultiAugs outperforms some burdensome transformer-based methods (Yang et al., 2021; Li et al., 2021b; Yuan et al., 2021), which reveals the significance of rational utilization of unlabeled data and advanced SSL techniques.

Table 4: Results on the COCO *val-set* with using the labeled *train-set* and unlabeled *wild-set* for training.

| Method | Backbone | Nets | AP | $AP_{.5}$ | AR | $AR_{.5}$ |
|---|---|---|---|---|---|---|
| Supervised (Xiao et al., 2018) | ResNet50 | Single | 70.9 | 91.4 | 74.2 | 92.3 |
| PoseDual (Xie et al., 2021) | ResNet50 | Dual | 73.9 | 92.5 | 77.0 | 93.5 |
| SSPCM (Huang et al., 2023) | ResNet50 | Triple | 74.2 | 92.7 | 77.2 | 93.8 |
| Pseudo-HMs (Yu et al., 2024) | ResNet50 | Dual | 74.1 | — | — | — |
| Ours (Single) | ResNet50 | Single | 74.4 | **93.6** | 77.4 | **94.0** |
| Ours (Dual) | ResNet50 | Dual | **74.6** | 93.5 | **77.6** | **94.0** |
| Supervised (Xiao et al., 2018) | ResNet101 | Single | 72.5 | 92.5 | 75.6 | 93.1 |
| PoseDual (Xie et al., 2021) | ResNet101 | Dual | 75.3 | 93.6 | 78.2 | 94.1 |
| SSPCM (Huang et al., 2023) | ResNet101 | Triple | 75.5 | **93.8** | 78.4 | 94.2 |
| Pseudo-HMs (Yu et al., 2024) | ResNet101 | Dual | 75.7 | — | — | — |
| Ours (Single) | ResNet101 | Single | 75.8 | 93.5 | 78.8 | 94.4 |
| Ours (Dual) | ResNet101 | Dual | **76.4** | 93.6 | **79.3** | **94.7** |
| Supervised (Xiao et al., 2018) | HRNet-w48 | Single | 77.2 | 93.5 | 79.9 | 94.1 |
| PoseDual (Xie et al., 2021) | HRNet-w48 | Dual | 79.2 | 94.6 | 81.7 | 95.1 |
| SSPCM (Huang et al., 2023) | HRNet-w48 | Triple | 79.4 | **94.8** | 81.9 | **95.2** |
| Pseudo-HMs (Yu et al., 2024) | HRNet-w48 | Dual | 79.4 | — | — | — |
| Ours (Single) | HRNet-w48 | Single | 79.3 | 94.6 | 81.9 | 95.1 |
| Ours (Dual) | HRNet-w48 | Dual | **79.5** | 94.6 | **82.1** | **95.2** |

**S4**: For the MPII dataset, we allocate its *train-set* as the labeled set and whole AIC as the unlabeled set. This is a more realistic setting where labeled and unlabeled images are from different datasets. The Tab. 7 (see Sec. A.2) shows results on the MPII *val-set*. Our method outperforms both the fully supervised HRNet (Sun et al., 2019) and semi-supervised PoseDual (Xie et al., 2021) by a large margin under the same backbone. It is worth noting that our semi-supervised MultiAugs with applying the model ensemble can even approach the supervised HRNet with using extra labeled AIC.

Table 5: Comparison to the SOTA methods on the COCO *test-dev*. The person detection results are provided by SimpleBaseline (Xiao et al., 2018) and flipping strategy is used.

| Method | Backbone | Input Size | Gflops | Params | AP | AR |
|---|---|---|---|---|---|---|
| SimpleBaseline (Xiao et al., 2018) | ResNet50 | 256×192 | 8.9 | 34.0 | 70.2 | 75.8 |
| HRNet (Sun et al., 2019) | HRNet-w48 | 384×288 | 32.9 | 63.6 | 75.5 | 80.5 |
| MSPN (Li et al., 2019) | ResNet50 | 384×288 | 58.7 | 71.9 | 76.1 | 81.6 |
| DARK (Zhang et al., 2020) | HRNet-w48 | 384×288 | 32.9 | 63.6 | 76.2 | 81.1 |
| UDP (Huang et al., 2020) | HRNet-w48 | 384×288 | 33.0 | 63.8 | 76.5 | 81.6 |
| TransPose-H-A6 (Yang et al., 2021) | HRNet-w48 | 256×192 | 21.8 | 17.5 | 75.0 | — |
| TokenPose-L/D24 (Li et al., 2021b) | HRNet-w48 | 384×288 | 22.1 | 29.8 | 75.9 | 80.8 |
| HRFormer (Yuan et al., 2021) | HRFormer-B | 384×288 | 26.8 | 43.2 | 76.2 | 81.2 |
| ViTPose (Xu et al., 2022) | ViT-Large | 256×192 | 59.8 | 307.0 | 77.3 | 82.4 |
| DUAL (+HRNet) (Xie et al., 2021) | HRNet-w48 | 384×288 | 65.8 | 127.2 | 76.7 | 81.8 |
| DUAL (+DARK) (Xie et al., 2021) | HRNet-w48 | 384×288 | 65.8 | 127.2 | 77.2 | 82.2 |
| SSPCM (+DARK) (Huang et al., 2023) | HRNet-w48 | 384×288 | 98.7 | 190.8 | **77.5** | **82.4** |
| Ours (Dual) (+HRNet) | HRNet-w48 | 384×288 | 65.8 | 127.2 | 76.8 | 81.8 |
| Ours (Dual) (+DARK) | HRNet-w48 | 384×288 | 65.8 | 127.2 | 77.3 | 82.3 |

**S5**: Finally, as shown in Tab. 8 (see Sec. A.2), our results on the MPII *test-set* surpass all those of previous fully supervised methods and two semi-supervised counterparts PoseDual and SSPCM. This further validates the effectiveness and superiority of our proposed method.

## 5.3 ABLATION STUDIES

For empirical studies in Fig. 2, 4, 5 and Tab. 1, 2, we conducted them follow the setting of COCO 1K with a total of 30 epochs. The learning rate drops $10\times$ twice separately at epochs 20 and 25. All studied models take ResNet18 as the backbone, and use the Single-Network framework. With the same setting, we conducted more ablation experiments to further analyze the proposed MultiAugs.

**Comparing to Other Advanced Augmentations** Three advanced augmentations are selected: RandAugment (Cubuk et al., 2020), TrivialAugment (Müller & Hutter, 2021) and YOCO (Han et al., 2022). We refer to them as $T_{RA}$, $T_{TA}$ and $T_{YOCO}$. For $T_{YOCO}$, it may be based on $T_{RA}$ or $T_{TA}$. And we compare them with previous SOTA augmentations $T_{JO}$ and $T_{JC}$ for SSHPE, and our recommended $T_{JOCO}$ and $T_{JCCM}$. We did not compare to Mixup families (Zhang et al., 2018; Hendrycks et al., 2020; Pinto et al., 2022; Liu et al., 2022) or AutoAug families (Cubuk et al., 2018; Lim et al., 2019; Hataya et al., 2020; Zheng et al., 2022). Because Mixup is verified not to work with SSHPE, and AutoAug needs to search optimal parameters. Finally, as shown in Tab. 6 and Fig. 8, our optimal combinations are always better than $T_{RA}$-based or $T_{TA}$-based $T_{YOCO}$, which are meticulously designed and also composed of existing basic augmentations. These further prove the superiority and conciseness of our synergistic combinations.

Table 6: Best mAP results of different augmentations on COCO *val-set*.

| Augmentations | Year | mAP |
|---|---|---|
| $T_{RA}$ (Cubuk et al., 2020) | CVPRW'2020 | 41.9 |
| $T_{TA}$ (Müller & Hutter, 2021) | ICCV'2021 | 40.2 |
| $T_{YOCO}$ (Han et al., 2022) $(T_{RA})$ | ICML'2022 | 42.5 |
| $T_{YOCO}$ (Han et al., 2022) $(T_{TA})$ | ICML'2022 | 41.6 |
| $T_{JC}$ (Xie et al., 2021) | ICCV'2021 | 42.4 |
| $T_{JO}$ (Huang et al., 2023) | CVPR'2023 | 41.9 |
| Ours $(T_{JCCM})$ | — | 42.7 |
| Ours $(T_{JOCO})$ | — | 43.7 |

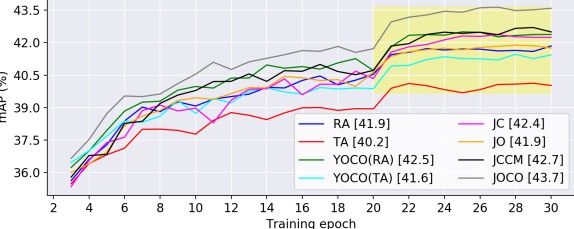

Figure 8: The convergence curves and best mAP results of different strong augmentations.

**Training Techniques of Multiple Heatmaps** In this part, we present additional techniques commonly used for unsupervised consistency training. Especially, for predicted multi-heatmaps, we propose to optimize them by applying the multi-loss learning (ML) as in Eq. 4. Other two alternative techniques are *confidence masking* (CM) and *heatmaps fusion* (HF). For CM, the consistency loss term in each mini-batch is computed only on keypoint channels whose maximum activation value is greater than a threshold $\tau$, which is set as 0.5. For HF, also termed as heatmaps ensemble, we sum and average multi-path heatmaps to obtain a fused heatmap for loss computing. Then, we compare MultiAugs (e.g., $m_{11}$ and $m_{12}$) using either of these three techniques ML, CM and HF.

As shown in Fig. 9, our ML is strictly superior than the other two techniques under either $m_{11}$ or $m_{12}$. For CM, we assume it may filter out some keypoint heatmaps with low confidence but high quality. This surely has a negative impact. For HF, although it is widely used in other SSL tasks for model ensemble, it may not necessarily be applicable to our intermediate keypoint heatmaps. We deem this is because each predicted heatmap is distinctive and meaningful (see Fig. 10). It is tricky to replace them equivalently with a fused heatmap. In comparison, our multi-loss learning is a simple yet effective choice.

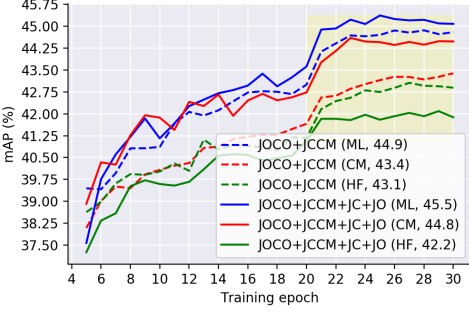

Figure 9: The convergence curves and best mAP results of two MultiAugs schemes with using three different training techniques.

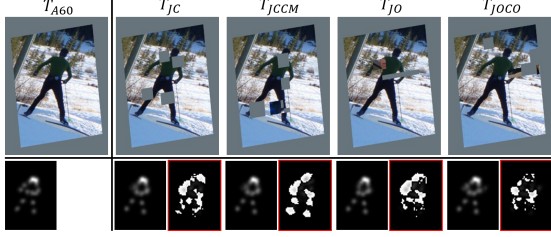

Figure 10: The predicted heatmaps of one easy augmentation ($T_{A60}$) and four hard augmentations in $m_{10}$. We also report the pixel-wise heatmap difference (with red borders) of each easy-hard pair to highlight subtle dissimilarities.

## 6 CONCLUSIONS

In this paper, we aim to boost semi-supervised human pose estimation (SSHPE) from two perspectives: data augmentation and consistency training. Instead of inventing advanced augmentations in isolation, we attempt to synergistically utilize existing augmentations, and handily generate superior ones by novel combination paradigms. The discovered collaborative combinations have intuitive interpretability. We verified their advantages in solving the SSHPE problem. For consistency training, we abandon the convention of stacking networks to increase unsupervised losses, and train a single network by optimizing multi-path consistency losses for the same batch of unlabeled images. Combined with the optimal hard augmentations set, this plain and compact strategy is proven to be effective, and leads to better performance on public benchmarks. Last but not least, we declare that the synergy effects of augmentations and multiple consistency losses are generic and generalizable for other SSL vision tasks such as image classification, object detection and semantic segmentation. We will explore them in the future.

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

# A  APPENDIX

## A.1  ANALYSIS OF SUPERIOR AUGMENTATIONS

In this part, we tentatively analyze why employing a superior augmentation to strongly augment the unlabeled data can improve model performance. Different from UDA (Xie et al., 2020a) using the improved connectivity of constructed graphs to explain, we start from the perspective of shaped feature space of unlabeled data based on the singular value spectrum, which is widely considered to be related to the model transferability and generalization (Chen et al., 2019; Xue et al., 2022). Specifically, we perform singular value decomposition (SVD) on features $\mathbf{F} \in \mathbb{R}^{N \times D 1}$ extracted by various trained models with different strong augmentations on one dataset: $\mathbf{F} = \mathbf{U}\boldsymbol{\Sigma}\mathbf{V}^T$, where $\mathbf{U}$ and $\mathbf{V}$ is the left and right singular vector matrices, respectively, and $\boldsymbol{\Sigma}$ denotes the diagonal singular value matrix $\{\sigma_1, \sigma_2, ..., \sigma_D\}$. Then, we plot cal-

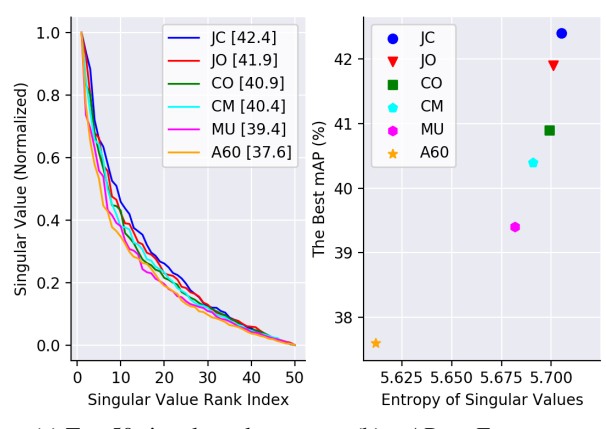

(a) Top-50 singular values  (b) mAP *vs.* Entropy

Figure 11: SVD analysis. The backbone of all six models is ResNet18. The 512-D features of 6,352 samples in COCO *val-set* are extracted.

culated singular values in Fig. 11a. To further measure the flatness of the singular value distribution, we calculate the entropy of normalized singular values $\mathsf{H}_{nsv}$:

$$\mathsf{H}_{nsv} = -\sum_{i=1}^{D} \frac{\sigma_i}{\sum_{j=1}^{D} \sigma_j} \log \frac{\sigma_i}{\sum_{j=1}^{D} \sigma_j}. \tag{7}$$

Usually, a larger $\mathsf{H}_{nsv}$ indicates that the feature space captures more structure in the data and thus spans more dimensions due to more discriminated representations learned. As shown in Fig. 11b, the model performance is *positively correlated* with the $\mathsf{H}_{nsv}$ value. Therefore, a superior augmentation facilitates better model generalization to unseen test sets.

Table 7: Results on the *val-set* of MPII dataset. HRNet is trained only on the MPII *train-set*. The "*" means using extra labeled dataset AIC. The "+" means applying the model ensemble.

| Method | Hea | Sho | Elb | Wri | Hip | Kne | Ank | Total |
|---|---|---|---|---|---|---|---|---|
| HRNet (Sun et al., 2019) | 97.0 | 95.7 | 89.4 | 85.6 | 87.7 | 85.8 | 82.0 | 89.5 |
| HRNet* (Sun et al., 2019) | 97.4 | 96.7 | 92.1 | 88.4 | 90.8 | 88.6 | 85.0 | **91.7** |
| PoseDual (Xie et al., 2021) | 97.4 | 96.6 | 91.8 | 87.5 | 89.6 | 87.6 | 83.8 | 91.1 |
| Ours (Dual) | 97.3 | 96.8 | 91.7 | 87.5 | 90.3 | 88.6 | 84.6 | 91.4 |
| Ours+ (Dual) | 97.3 | 96.8 | 91.9 | 88.1 | 90.6 | 89.2 | 85.0 | **91.7** |

Table 8: Comparisons on the *test-set* of the MPII dataset. We use HRNet-w32 as the backbone. The input image size is 256×256. The MPII (w/ labels) and AIC (w/o labels) are used for SSL training.

| Method | Hea | Sho | Elb | Wri | Hip | Kne | Ank | Total |
|---|---|---|---|---|---|---|---|---|
| Newell et al. (2016) | 98.2 | 96.3 | 91.2 | 87.1 | 90.1 | 87.4 | 83.6 | 90.9 |
| Xiao et al. (2018) | 98.5 | 96.6 | 91.9 | 87.6 | 91.1 | 88.1 | 84.1 | 91.5 |
| Ke et al. (2018) | 98.5 | 96.8 | 92.7 | 88.4 | 90.6 | 89.4 | 86.3 | 92.1 |
| Sun et al. (2019) | 98.6 | 96.9 | 92.8 | 89.0 | 91.5 | 89.0 | 85.7 | 92.3 |
| Zhang et al. (2019) | 98.6 | 97.0 | 92.8 | 88.8 | 91.7 | 89.8 | 86.6 | 92.5 |
| Xie et al. (2021) | 98.7 | 97.3 | 93.7 | 90.2 | 92.0 | 90.3 | 86.5 | 93.0 |
| Huang et al. (2023) | 98.7 | 97.5 | 94.0 | **90.6** | **92.5** | **91.1** | 87.1 | 93.3 |
| Ours (Dual) | **98.8** | **97.6** | **94.1** | 90.3 | 92.4 | **91.1** | **87.2** | **93.4** |

---

[1]We denote $N$ as the number of samples and $D$ as feature dimensions (*a.k.a*, $D \le N$).

## A.2 MORE PERFORMANCE COMPARISON DETAILS

In this section, we place more quantitative data that do not have space to present in the main content, including the detailed results of MPII *val-set* and MPII *test-set* in Tab. 7 and Tab. 8, respectively.

Besides, to further verify the effectiveness of our method, we conducted experiments on an indoor overhead fisheye human keypoint dataset WEPDTOF-Pose which is based on CEPDOF (Duan et al., 2020) and WEPDTOF (Tezcan et al., 2022). Followed SSPCM (Huang et al., 2023), we used the complete WEPDTOF-Pose *train-set* (4,688 person instances) as the labeled dataset, and CEPDOF (Duan et al., 2020) with 11,878 person instances as the unlabeled dataset for experiment. The WEPDTOF-Pose *test-set* (1,179 person instances) is used as the evaluation set. The metric of mAP (Lin et al., 2014) is reported for comparing. More details of WEPDTOF-Pose dataset can be found in (Huang et al., 2023). It should be noted that SSPCM does not open source the BKFisheye dataset included in WEPDTOF-Pose, so we cannot conduct corresponding experimental comparisons involving the BKFisheye with it.

For training on WEPDTOF-Pose, the hard augmentations set $\widehat{\mathcal{T}}^4 = \{T_{JOCO}, T_{JCCM}, T_{JC}, T_{JO}\}$ is chosen as in settings **S1**, **S2** and **S3**. The used backbone is ResNet-18. Given the particularity of the fisheye dataset, the random rotation range used in all hard data augmentations is (-90°, 90°), which means $T_{A60}$ is changed into $T_{A90}$. We use the Adam optimizer to train these models. The initial learning rate is 1e-3, which decreases to 1e-4 and 1e-5 at 140 epochs and 180 epochs, respectively, with a total of 200 epochs. As shown in Tab. 9, our method always achieves the best AP and AR results whether using a single network or a dual network structure, surpassing the previous SOTA method SSPCM using a triple network. These experiments in the fisheye domain once again demonstrate the superiority and universality of our proposed MultiAugs.

Table 9: Comparison of our method to the SOTA methods on the dataset WEPDTOF-Pose collected by indoor overhead fisheye camera.

| Method | Network Number | Labeled Dataset | Unlabeled Dataset | AP | AR |
|---|---|---|---|---|---|
| Supervised (Xiao et al., 2018) | 1 | WEPDTOF-Pose | — | 49.5 | 53.4 |
| PoseCons (Xie et al., 2021) | 1 | WEPDTOF-Pose | CEPDOF | 54.6 | 58.1 |
| PoseDual (Xie et al., 2021) | 2 | WEPDTOF-Pose | CEPDOF | 55.1 | 59.0 |
| SSPCM (Huang et al., 2023) | 3 | WEPDTOF-Pose | CEPDOF | 55.6 | 60.0 |
| Ours (Single) | 1 | WEPDTOF-Pose | CEPDOF | 56.5 | 60.6 |
| Ours (Dual) | 2 | WEPDTOF-Pose | CEPDOF | **57.1** | **61.3** |

In addition, we consider that reporting the comparison results based on the more important ResNet-50 is more convincing than ResNet-18. Therefore, we replaced the backbone in Tab. 3 with ResNet-50 according to setting **S1** and re-conducted the comparative experiments. The results are shown in Tab. 10. Similar to using ResNet-18, our method can still achieve a clear advantage. When using a single network, our method outperforms PoseCons and Posedual, while being comparable to SSPCM. And our dual-network based approach achieves significant advantages.

Table 10: AP of different methods on COCO *val-set* when different numbers of labels are used. The backbone of all methods is ResNet-50. The trend of accuracy change is shown in Fig. 12.

| Method | Net. Num. | 1K | 5K | 10K | All |
|---|---|---|---|---|---|
| Supervised (Xiao et al., 2018) | 1 | 34.8 | 50.6 | 56.4 | 70.9 |
| PoseCons (Xie et al., 2021) | 1 | 43.1 | 57.2 | 61.8 | — |
| PoseDual (Xie et al., 2021) | 2 | 48.2 | 61.1 | 65.0 | — |
| SSPCM (Huang et al., 2023) | 3 | 49.8 | 61.8 | 65.5 | — |
| Ours (Single) | 1 | 49.3 | 61.4 | 65.2 | — |
| Ours (Dual) | 2 | **51.7** | **62.9** | **66.3** | — |

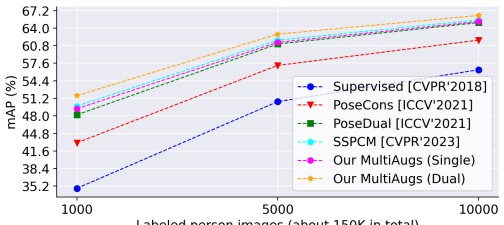

Figure 12: Comparison between state-of-the-art SSHPE methods and our proposed Multi-Augs on the COCO dataset.

Moreover, we deem that conclusions drawn from testing and evaluation on small-scale data may not necessarily be generalized to other datasets. Therefore, we repeated the comparison in setting **S1** and Tab. 3 by replacing the COCO dataset into MPII dataset. Specifically, we conducted experiments using the first 1K samples as labeled data and the left 39K samples as unlabeled data in MPII. The

validation set of MPII is used to evaluate. The backbone is ResNet-18. The final comparison results are shown in Tab. 11. Not surprisingly, our method still maintains a clear lead in performance, both in terms of overall accuracy and the specific accuracy of each joint. These experiments once again demonstrate that our method is indeed universally effective and superior across different datasets.

Table 11: Results on the *val-set* of MPII dataset. All models utilize ResNet-18 as the backbone. The best results are highlighted in bold.

| Method | Hea | Sho | Elb | Wri | Hip | Kne | Ank | Total |
|---|---|---|---|---|---|---|---|---|
| Supervised (Xiao et al., 2018) | 89.6 | 84.8 | 72.0 | 58.4 | 57.8 | 49.4 | 41.2 | 65.3 |
| PoseCons (Xie et al., 2021) | 92.7 | 87.6 | 74.5 | 67.9 | 72.3 | 64.2 | 59.4 | 75.2 |
| PoseDual (Xie et al., 2021) | 93.3 | 88.4 | 75.0 | 67.3 | 72.6 | 65.3 | 59.7 | 75.6 |
| SSPCM (Huang et al., 2023) | 93.5 | 90.6 | 80.2 | 71.3 | 75.9 | 68.9 | 62.3 | 78.3 |
| Ours (Single) | 94.1 | 91.1 | 80.5 | 72.2 | 76.3 | 69.2 | 62.8 | 79.1 |
| Ours (Dual) | **94.7** | **92.4** | **81.2** | **73.3** | **76.8** | **70.6** | **63.9** | **79.7** |

## A.3 QUALITATIVE VISUALIZATION COMPARISON

To make our advantages more intuitively demonstrated, we have added qualitative visualization comparison results, mainly including the conventional human images from the COCO val-set and the fisheye camera images from the WEPDTOF-Pose dataset. We take use of the backbone ResNet-18 for all compared methods to highlight the their performance differences. For models trained on COCO dataset, we use the label set with 10K samples for comparison. As shown in Fig. 13, pure supervised learning methods usually are prone to making mistakes or messing up, and other SSHPE methods do not perform well in some occlusion cases or edge keypoint detection. While, our method often obtains more accurate estimations.

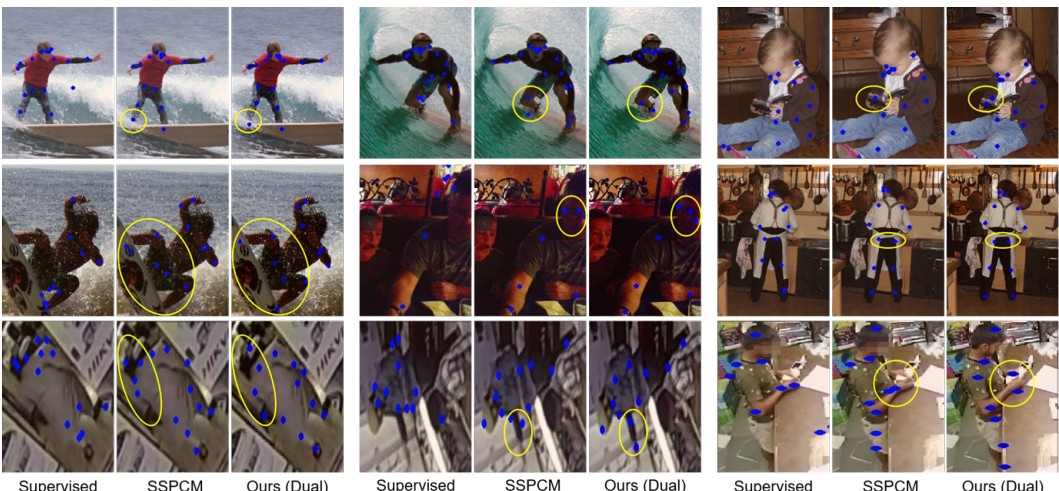

Figure 13: Qualitative results on COCO val-set (the first 6 examples) and WEPDTOF-Pose test-set (the last 3 examples). The predicted results of methods Supervised and SSPCM are directly fetched from the supplementary paper of SSPCM (Huang et al., 2023). The details of the comparison between SSPCM and our predictions are highlighted in yellow circles for quick identification.

## A.4 PARAMETERS OF BASIC AUGMENTATIONS

The hyper-parameters involved in each augmentation are indeed important. In order to make a fair comparison, each basic augmentation we selected is derived from various compared methods without additional fine-tuning. For example, the parameters of Joint Cutout are the same as those in PoseDual (Xie et al., 2021) which used JC5, and the parameters of Joint Cut-Occlude are the same as those in SSPCM (Huang et al., 2023) which used JO2. We list the parameters of the basic augmentations used in this paper in Tab. 12, so that readers can quickly and clearly know these details. Please see Fig. 3 for some visualization results after applying augmentations.

Table 12: The hyper-parameters details of the basic augmentations.

| Aug. | Type | Description |
|------|------|-------------|
| $T_{A30}$ | easy | random scale within range $[0.75, 1.25]$, random rotation within range $(-30°, 30°)$. |
| $T_{A60}$ | hard | random scale within range $[0.75, 1.25]$, random rotation within range $(-60°, 60°)$. |
| $T_{CO}$ | hard | random generation **5** zero value patches with size $20 \times 20$. |
| $T_{CM}$ | hard | random cropping **2** patches from other images with size $20 \times 20$. |
| $T_{JC}$ | hard | random generation **5** zero value patches with size $20 \times 20$ around predicted joints. |
| $T_{JO}$ | hard | random cropping **2** patches from other images with size $20 \times 20$ around predicted joints. |

## A.5 ADDITIONAL ABLATION STUDIES

Firstly, we need to investigate whether the multi-path consistency loss strategy is sensitive to the training batch size. In fact, when designing the ablation studies in Tab. 1 using the single-path loss and Tab. 2 using multi-path losses, we always chose a fixed batch size 32 to perform all experiments. Moreover, in final comparative experiments (see Tab. 3), we still keep the batch size as 32 and use the optimal four-path losses. Now, in order to investigate the possible impact of different batch sizes, we report the effects of PoseCons and PoseDual when the batch size is 128. As can be seen in Tab. 13, after increasing the batch size of PoseCons or PoseDual accordingly, the final mAP results under different labeling rates (e.g., 1K, 5K and 10K) did not get significantly better. This indicates that batch size does not have a large impact on the performance of existing methods.

Then, we also need to conduct additional experiments to probe whether to use a single constant easy augmentation as input for multi-path losses (the pair $\{I_e\} + \{I_{h_1}, ..., I_{h_n}\}$, termed as 1-vs-n) or to use different easy augmentations multiple times as input (the pair $\{I_{e_1}, ..., I_{e_n}\} + \{I_{h_1}, ..., I_{h_n}\}$, termed as n-vs-n). As shown in Tab. 14, whether using 1-v-n augmented input or n-vs-n augmented input, the final mAP results obtained under various labeling rates are not significantly different. This is mainly because the used easy augmentation is always fixed (e.g., $T_{A30}$), so the input does not change in essence when applying 1-vs-n input or n-vs-n.

Table 13: AP results of baseline methods PoseCons and PoseDual after increasing the batch size. The used backbone is ResNet-18.

| Method | Net. Num. | Loss Num. | Batch Size | 1K | 5K | 10K |
|--------|-----------|-----------|------------|------|------|------|
| PoseCons | 1 | 1 | 32 | 42.1 | 52.3 | 57.3 |
| PoseCons | 1 | 1 | 128 | 42.3 | 52.6 | 57.5 |
| PoseDual | 2 | 1 | 32 | 44.6 | 55.6 | 59.6 |
| PoseDual | 2 | 1 | 128 | 44.9 | 58.7 | 59.6 |
| Ours (Single) | 1 | 4 | 32 | 45.5 | 56.2 | 59.9 |
| Ours (Dual) | 2 | 4 | 32 | 49.7 | 58.8 | 61.8 |

Table 14: AP results of our methods based on single-network or dual-network after adjusting the augmentation way of inputs. The used backbone is ResNet-18.

| Method | Net. Num. | Input | 1K | 5K | 10K |
|--------|-----------|-------|------|------|------|
| Ours (Single) | 1 | 1-vs-n | 45.5 | 56.2 | 59.9 |
| Ours (Single) | 1 | n-vs-n | 45.6 | 56.4 | 59.8 |
| Ours (Dual) | 2 | 1-vs-n | 49.7 | 58.8 | 61.8 |
| Ours (Dual) | 2 | n-vs-n | 49.7 | 58.9 | 61.9 |

Finally, in order to fairly and reasonably reflect the efficiency of our method in the training phase, we follow the setting **S1** (using ResNet-18 as the backbone, batch size is set to 32, total training epochs are 30, and the amount of labeled data is 1K), and conduct each experiment on four 3090 graphics cards (with each containing 24 GB memory) to compare the training time of our method with that of PoseCons and PoseDual. The strong augmentation used by PoseCons or PoseDual is $T_{JC}$. Considering that our method often uses different strong augmentations, their computation is not the main bottleneck. Therefore, in order to be fair, all strong augmentations in our method are also replaced into $T_{JC}$. Assuming that the total training time of PoseCons is one unit time $T_0$, which is actually about 7 hours. Then the total training time of running other methods is summarized in Tab. 15.

Table 15: Quantitative comparison of training time between baseline methods (PoseCons and PoseDual) and our proposed method. The integer number with the marker # in our method means how many multi-path losses are used.

| Method | PoseCons | Ours (Single,2#) | Ours (Single,3#) | Ours (Single,4#) | PoseDual | Ours (Dual,2#) | Ours (Dual,3#) | Ours (Dual,4#) |
|--------|----------|------------------|------------------|------------------|----------|----------------|----------------|----------------|
| Time | $T_0$ | $1.36*T_0$ | $1.50*T_0$ | $1.83*T_0$ | $2.49*T_0$ | $2.62*T_0$ | $2.88*T_0$ | $3.14*T_0$ |

From those results in Tab. 15, we can see that when using four-path losses, although the training time increases, it is still faster than PoseDual ($1.83*T_0$ vs. $2.49*T_0$ ). Referring to the quantitative results in Tab. 3 of the main paper, our method based on single-network using four-path losses achieves higher mAP than PoseDual. In addition, when using dual networks with four-path losses, the total training time does not increase significantly ($2.49*T_0$ vs. $3.14*T_0$ ). These indicate that our method is both efficient and effective.

