# OpenReview forum: "Boosting Semi-Supervised 2D Human Pose Estimation by Revisiting Data Augmentation and Consistency Training"
_ICLR.cc/2025/Conference — Submitted to ICLR 2025_

### Official Review · Reviewer_z46M · 2024-10-30

**Soundness:** 2
**Presentation:** 2
**Contribution:** 2
**Rating:** 3
**Confidence:** 4

**Summary:**

The paper proposes a data augmentation method with the consistency training strategy to improve the performance of semi-supervised 2D human pose estimation,

**Strengths:**

1.This paper identifies that the existing SSHPE methods lack rigor in ranking the difficulty levels of applied data augmentations and discovers synergistic effects among different augmentations. It proposes a more rigorous difficulty ranking for data augmentations.
2.This paper provides a comprehensive evaluation of existing advanced data augmentation methods. Rather than designing new augmentation techniques, the paper employs rule-based constraints to combine existing augmentations, nominating the most likely superior combinations: TJOCO and TJCCM.
3.A novel multi-path approach that applies multi-path augmentations and multi-path loss training to a single network can surpass certain dual network.

**Weaknesses:**

1.The introduction of new data augmentation requires a re-evaluation of the selection of the optimal augmentation combination.
2.The combined data augmentation will increase the training time. Is this time consumption still lower than that of stacked networks?
3.Different combinations may yield varying performance across datasets, so can the selected optimal combination consistently provide the best results?
4.The selection process for the optimal augmentation combination is time-intensive.

5. Although the paper provides a ranking of data augmentation techniques, the criteria for evaluating the difficulty levels of augmentations appear somewhat heuristic and may lack a strong quantitative foundation.

6. The paper lacks experiments using only the multi-path loss, making it difficult to determine the individual performance improvements contributed by the multi-path loss and the two new data augmentations.

7. The paper does not clearly explain the rationale for choosing the TJOCO and TJCCM data augmentations.

**Questions:**

Please see the weakness part.

---

> ### Author Response · Authors · 2024-11-19
> **Official Responses by Authors**
>
> Dear Reviewer z46M,
>
> Thank you for your inspiring affirmation, detailed review and insightful queries regarding our paper. Your feedback is invaluable in improving our work. Below, we address each of your points to clarify our methodology and findings.
>
> ***
> ***W1: What if a new augmentation is introduced?***
>
> **R1:** We acknowledge that for a newly added basic augmentation, some evaluation is still required to select possible advanced augmentation combinations. However, as described in the previous responses to W3 raised by `Reviewer 4hx9` and W1 raised by `Reviewer r3Sz`, thanks to the proposed paradigm for generating superior augmentations in this paper, we can simplify and speed up this process, requiring only a small amount of workload.
>
> Specifically, for the newly added basic augmentations, we first need to evaluate and rank their difficulty to get a preliminary understanding of their effects and usability. Then, we can roughly judge which augmentations are mutually exclusive or synergistic based on experience. For example, occlusion of keypoints (e.g., $T_{JC}$ and $T_{JO}$) and randomly generated occlusion (e.g., $T_{CO}$ and $T_{CM}$) are essentially synergistic. We can also roughly classify and filter through singular value decomposition (SVD) analysis, such as that shown in Figure 11. After that, we will confidently recommend new and stronger augmentation combinations based on the three concise criteria proposed in Section 3.1, including not using MixUp, not repeating with the same type of augmentation, and not stacking too many augmentations.
>
> Finally, let's give an additional example. For instance, in Section 5.1, we introduced a new basic augmentation, YOCO (You Only Cut Once), which is based on other existing augmentations such as RandAugment or TrivialAugment. We have actually evaluated its basic effect in Table 6, which is roughly the same difficulty as $T_{JC}$ or $T_{JO}$. Then, considering that YOCO's operation is to crop and re-stitch the same image, it is a completely new type in form. Therefore, it is easy to think that it can be combined with the strongest augmentation $T_{JOCO}$ or $T_{JCCM}$ that we have already introduced, that is, to get a better combination $T_{JOCO+YOCO}$ or $T_{JCCM+YOCO}$. The process is not very time-consuming or difficult to understand, so it has the potential and value to be widely promoted for unsupervised data augmentation generation.
>
> ***
> ***W2: Training time after using augmentations combination***
>
> **R2:** Thanks for pointing out this important detail. In fact, the combination of different augmentations will eventually result in one augmentation after continuous execution, and will not significantly increase the overall time. The total time consumption is related to the basic augmentation itself. For example, the time consumption of $T_{JCCM}$ or $T_{JOCO}$ is similar to that of $T_{JC}$ or $T_{JO}$.
>
> The design that actually increases the training time is another strategy we proposed, multi-path consistency loss. We have quantitatively calculated and compared the time cost of different number of path augmentations in our reply to Q2 raised by `Reviewer r3Sz`. Please refer to it for more details. In short, the final conclusion is that the design of multi-path loss will not significantly prolong the training time, but will bring great performance improvement. It is a simple and efficient strategy for addressing the SSHPE problem.
>
> ***
> ***W3: Stability of optimal augmentations combination***
>
> **R3:** This is also an important issue. In our response to W5 raised by `Reviewer r3Sz`, we newly added a detailed comparative experiment on the MPII dataset following the setting S1, and the trend of the final results is consistent with the conclusions on the COCO dataset in Table 3. Our method still achieved clear advantages whether using a single network or dual networks. We have updated these results in our revised paper in **Appendix A.2**. These further prove that the optimal augmentation combination and multi-path consistency loss we proposed are better and can continue to achieve leading results across different datasets.
>
> ***
> **Continue in the next comment.**

---

> ### Author Response · Authors · 2024-11-19
> **Official Responses by Authors**
>
> ***W4: Selection process of augmentations combination***
>
> **R4:** In fact, in the field of semi-supervised learning, it is not easy to discover or invent new advanced augmentations. We can find that there are still a lot of research exploring this issue to date, please refer to the third paragraph of Section 2 and the second paragraph of Section 5.3 of the main text. Under this consensus, this paper attempts to propose a standardized paradigm to produce strong augmentations, rather than just proposing a new one and ending it. Our approach is to continue to use the new basic augmentations proposed by the community to further obtain a better one. Therefore, although the advanced augmentation generation process described in Chapter 3.1 seems complicated, it provides a feasible route and can help save trial and error costs in practice. If we can eventually find superior augmentation at this cost, we think it is worthwhile and meaningful.
>
> ***
> ***W5: Quantitative analysis of augmentation difficulty ranking***
>
> **R5:** This is indeed an issue worth exploring in depth. As pointed out in our paper, we initially found that the approach in PoseDual was to augment the images directly on the test set. Then, in order to reflect the difficulty of different augmentations, they evaluate the same model when facing different types of augmented inputs, and record the severity of the drop in accuracy indicators. The obvious drawback of this strategy is that if we directly input a noisy image, the final accuracy will be close to 0, but such augmentation is not desirable.
> Therefore, our approach of utilizing ablation experiments on the training set is more intuitive and convincing. Although it seems rather heuristic, the quantitative data comparison shown in Tables 1 and 2 provides us with indisputable assurance and confidence in proposing a new superior augmentation. We also confirmed in the experimental part that this paradigm is indeed effective.
>
> In addition, we also try to give a theoretical explanation for why different augmentations have different levels of difficulty in **Appendix A.1**. From the perspective of singular value decomposition (SVD), we find that the so-called advanced augmentation is more difficult because the average entropy of its singular values is higher, which means that after applying the corresponding augmentation, the augmented input may span a larger feature space, thereby helping to improve the generalization of the model. We hope that this explanation can help make up for the theoretical foundation of the proposed strategy in Section 3.1.
>
> ***
> ***W6: Ablation studies of the multi-path loss***
>
> **R6:** This is indeed an important concern. In fact, we have described in detail the effectiveness verification of the multi-path consistency loss strategy in the last two paragraphs of Section 3.2. For more detailed results, please refer to Table 2 and Figure 3. Among them, schemes $m_9$ and $m_{10}$ show the experiments of using the same strong augmentation to calculate two-path losses. After comprehensive comparison with scheme $m_{11}$, it can be found that different multi-path strong augmentations also have synergistic effects, which will perform better than using only a single type augmentation as in schemes $m9$ and $m_{10}$.
> In addition, by comparing some experiments in Table 1 and Table 2, such as $c_{10}$-$m_6$, $c_{11}$-$m_7$, $c_{12}$-$m_4$ and $c_{13}$-$m_5$, we can also find that multi-path loss itself has a positive effect, which can help alleviate the negative impact of excessive accumulation of multiple basic strong augmentations.
>
> ***
> ***W7: Reasons for selecting two optimal augmentations***
>
> **R7:** The reasons for choosing $T_{JOCO}$ and $T_{JCCM}$ can be found in the second and third paragraphs of Section 3.1. Specifically, first, we focus on the HPE task in this paper, and it is not beneficial to choose MixUp-related augmentations. For specific explanations, please refer to our response to W1 raised by `Reviewer 4hx9`. Secondly, considering the synergistic effects between different augmentations, we tend not to apply the same type of augmentation to the image. According to this criterion, we can exclude combinations $T_{COCM}$, $T_{JCCO}$, $T_{JOCM}$ and $T_{JOJC}$, leaving only $T_{JOCO}$ and $T_{JCCM}$. Finally, an intuitive idea is that we'd better not apply too many augmentations to an image, otherwise it may backfire and destroy the semantic information in the image. In extreme cases, it may directly make the image unrecognizable or meaningless. Therefore, we do not recommend stacking too many augmentations. These empirical conclusions or criteria are finally quantitatively verified in Table 1 and Figure 2.
>
> ***
> We hope these responses address your concerns and provide a clearer understanding of our approach and its capabilities. We are committed to making the necessary revisions to reflect these clarifications and enhance the overall quality of our paper.

---

> > ### Comment · Reviewer_z46M · 2024-11-26
> > **Rebuttal**
> >
> > Thanks for the efforts in the response. However, the rebuttal cannot address most of my concerns and I am willing to keep my final rating.

---

> > > ### Author Response · Authors · 2024-11-27
> > > **Official Responses by Authors**
> > >
> > > Thanks for your kind reply.
> > >
> > > In any case, thank you for your hard review efforts. We will do our best to further improve the current work.

---

### Official Review · Reviewer_r3Sz · 2024-10-31

**Soundness:** 3
**Presentation:** 3
**Contribution:** 2
**Rating:** 6
**Confidence:** 4

**Summary:**

The authors propose a method for boosting 2D human pose estimation performance in a semi-supervised setting. Towards this, the authors propose two cores: advanced data augmentation and concise consistency training ways. They hypothesize that these two cores help the model’s interpretability, benefitting from the augmentations, and aid it in performing superior to SoTA semi-supervised approaches.

**Strengths:**

1. The authors propose a simple and seemingly effective way to improve 2d human pose estimation from RGB images using advanced data augmentation combinations and a multi-path augmentation framework for training a single network.
2. Table 3: the proposed approach outperforms other baselines showing its effectiveness on the COCO val set with different labeled train set sizes.
3. Table 4: the proposed approach outperforms other baselines when using the Resnet50 and ResNet101 architectures showing its effectiveness on the COCO val set when using the entire COCO labeled train set with unlabeled wild set for training.
4. Table 9: Results on WEPDTOF-Pose which is an indoor dataset indicate that the proposed approach may be generalizable across datasets.

**Weaknesses:**

1. While the overall approach seems to work somewhat effectively, the contribution in the novelty aspect is limited. The authors propose a method to combine the different augmentations to create sensible hard augmentations but the takeaways are generalized in nature and seem obvious. For example, selecting combining augmentations in Sec 3.1 does not seem to take out-of-the-box thinking and may be deduced from running a combination of experiments.
2. Table 4: while the gains are noticeable when using ResNet as the backbone, the same cannot be said for HRNet as the backbone. This raises concerns regarding the effectiveness of using the proposed augmentations and approach. Can the authors justify why choosing a different architecture may not result in a noticeable performance boost? The current results indicate the proposed approach may be architecture-dependent.
3. Sec 5.2 S3 → is there a reason the authors do not train for 400 epochs if it is the only other thing besides the number of networks presenting their method from performing not as well as SSPCM?
4. Table 5: the proposed approach performs similarly to DUAL on the test set. This raises concerns over the generalizability of the method.
5. Sec 5.2 S1 → Superior performance on 1 dataset for a small number of samples may not necessarily generalize to using small train sets from other datasets. Is it possible to show results on other datasets, e.g. MPII (which is already evaluated), H3.6, or LSP? Additionally, can the authors justify this statement? The authors state in Dataset Setup that the small subsets of 1K, 5K, and 10K were randomly selected. From the reported numbers, the corresponding baselines do not seem to be trained on the same selected labeled set. It may then be likely (not certainly) that the reported numbers for the proposed approach may be biased towards the evaluation set because of the selected subset for training.
6. S5 Table 8: While the overall performance is better by 0.1 (not significant), the method outperforms 4 of 7 joints (as opposed to claiming all).

**Questions:**

1. The authors need to explain how they divide the different sections of Table 5. This is currently unclear from the table.
2. What is the effective training time of the proposed approach under different conditions/networks?
3. What is the significance of Fig.7 when Table 3 already presents the same results?

---

> ### Author Response · Authors · 2024-11-19
> **Official Responses by Authors**
>
> Dear Reviewer r3Sz,
>
> Thank you for your high appreciation, constructive feedback and insightful questions regarding our paper. We appreciate the opportunity to clarify the aspects you have highlighted. Please find our responses to your queries below.
>
> ***
> ***W1: Selection of augmentations combination***
>
> **R1:** First of all, thank you for your affirmation of the effectiveness of the method we proposed to obtain new advertisements. In fact, although it seems intuitive, there is no exactly the same work before us to try to propose and verify the combination synergy effect between multiple basic augmentations. A technical route that has a different starting point from our work but very similar final results is the AutoAug families (see the second paragraph in Section 5.3), which require offline search for optimal parameters of different augmentations and are usually highly dependent on the dataset utilized.
>
> Differently, we first ranked the difficulty of the existing basic augmentations, and then proposed three concise guidelines to recommend more advanced augmentation combinations, and used a large number of ablation experiments to verify these criteria. According to these criteria explained in Section 3.1, we do not have to verify all possible combinations one by one, and can rapidly be compatible with new basic augmentations, which provides a feasible pipeline for us to quickly obtain the optimal augmentation combination. For more details, we recommend reading our response to W3 raised by `Reviewer 4hx9`, which explains similar concerns.
>
> ***
> ***W2: Effects of different backbones***
>
> **R2:** Generally, the performance improvement in Table 4 is negatively correlated with the total number of parameters or basic capabilities of the backbone used. That is to say, for the same training and testing settings, using ResNet-50, ResNet-101 and HRNet-w48 as backbones respectively, the final performance will inevitably get better and better, regardless of whether full supervision or semi-supervision is used. Meanwhile, the predictable accuracy of the test set itself has an upper limit. This means that the higher the accuracy (here is mAP), the closer to performance saturation. Smaller absolute improvements are most likely related to the indicator limits of the evaluated dataset. Therefore, after using the most powerful HRNet-w48 as the backbone, the absolute improvement in mAP appears to be minor. As a more sensitive measure, we can calculate the relative performance improvement from SSPCM to Ours in each case.
>
> ```
> ---------------------------------------------------------------------------------
> Method		| Backbone	| Nets	| AP	| Abs.Imp.	| Rel.Imp.
> ---------------------------------------------------------------------------------
> Supervised	| ResNet50	| 1	| 70.9	| 0.0		| ---
> SSPCM		| ResNet50	| 3	| 74.2	| 3.3		| ---
> Ours (Dual)	| ResNet50	| 2	| 74.6	| 3.7		| 12.12%
> ---------------------------------------------------------------------------------
> Supervised	| ResNet101	| 1	| 72.5	| 0.0		| ---
> SSPCM		| ResNet101	| 3	| 75.5	| 3.0		| ---
> Ours (Dual)	| ResNet101	| 2	| 76.4	| 3.9		| 30.00%
> ---------------------------------------------------------------------------------
> Supervised	| HRNetw48	| 1	| 77.2	| 0.0		| ---
> SSPCM		| HRNetw48	| 3	| 79.4	| 2.2		| ---
> Ours (Dual)	| HRNetw48	| 2	| 79.5	| 2.3		| 4.55%
> ---------------------------------------------------------------------------------
> ```
>
> It can be found that although the absolute improvement based on HRNet is not obvious, our relative improvement cannot be ignored (about 5%). This shows that our method is not necessarily related to the network structure. In addition, we choose to use the current backbones for fair comparison with previous SOTA methods. Using more advanced backbones (such as vision transformers) to pursue higher accuracy/mAP is beyond the scope of this paper. However, our proposed method does not rule out the possibility of being applicable to other network structures, which will be left for future research.
>
> ***
> **Continue in the next comment.**

---

> > ### Comment · Reviewer_r3Sz · 2024-11-24
> > **Response to authors**
> >
> > Thank you for your response to my questions.
> >
> > 1. Response to W1R1: While I agree that "there is no exactly the same work before us to try to propose and verify the combination synergy effect between multiple basic augmentations", my point was that such a proposed method needs to run a set of optimal strategy experiments (similar to searching for optimal hyperparameters) and simply proposing a sequence of augmentation schemes (from existing ones) does not require formulating the problem in a novel sense. Furthermore, the gains are not significant enough to be considered an extraordinary contribution.
> >
> > For example, SimCLR [1] not only proposes the composition of augmentations but also shows how the overall proposed simple framework of contrastive learning gives highly noticeable performance gains.
> >
> >
> > 2. Response to W2R2: I believe it is unfair to base your comparison on a supervised approach in terms of relative improvement. A fair comparison would be a direct comparison to SSPCM (an SSHPE method) where it is evident that the gains are marginal (except ResNet101).
> >
> >
> > [1] Chen et al., "A Simple Framework for Contrastive Learning of Visual Representations"

---

> > > ### Author Response · Authors · 2024-11-25
> > > **Official Responses by Authors**
> > >
> > > Thanks for your kind reply.
> > >
> > > For W1R1, in fact, we have performed relevant ablation experiments in Table 1 and Figure 4 to verify the found optimal augmentation combinations. The related basic augmentations all used the same hyper-parameter settings as their respective sources to ensure fair comparison (details can be found in Appendix `A.4`). These quantitative results can provide evidence for our proposed intuitive principles.
> > >
> > > For W2R2, we tried to provide a comparative perspective to illustrate that it is indeed more difficult to achieve absolute performance improvement after the basic supervision network is strong enough. This actually also indirectly shows that it is not easy to clearly observe the performance differences of various methods under such settings. Therefore, we recommend that reviewers can refer to the significant advantages of our method over other SSHPE methods when the label annotation rate is scarce (please see Tables 3/9/10/11). This superiority is more valuable in practical applications, such as fisheye cameras, ego perspectives, low light conditions, etc. where human keypoint labels are not readily available in large quantities or are difficult to obtain.

---

> > > > ### Comment · Reviewer_r3Sz · 2024-11-30
> > > > **Response to Authors**
> > > >
> > > > ​​I thank the authors for their detailed responses. While most of my concerns are wholly or partially resolved, the major concern about proposing the solution or formulating the problem in a novel aspect as explained in my responses to authors remains. I have slightly raised my score based on the authors’ responses to experimental concerns.
> > > >
> > > > 1. W1R1: The concern is about the novelty of designing a method as opposed to running a set of experiments for hyperparameters search.
> > > >
> > > > 2. W3R3: While I understand that a code inference might help with the quantitative reproduction of results, it may not help in determining why their network performs better. As the authors claim to design an efficient network performing on par with other SShPE methods, the comparison has to be in accordance. The current performance does not reflect the claims and no clear explanation is available.
> > > >
> > > > 3. W2R2, W4R4, W5R5, and W6R6: The responses resolve my concern.

---

> > > > > ### Author Response · Authors · 2024-12-01
> > > > > **Official Responses by Authors**
> > > > >
> > > > > Thank you again for your hard work in reviewing and responsible discussions.
> > > > >
> > > > > There are indeed some parts of our work that need to be further improved, and in fact we are still in the process of continuous exploration and optimization. Compared with the comparison on the COCO or MPII keypoint dataset that is close to performance saturation, we prefer to generalize the design and findings of this paper to other similar SSL fields where data labels are more scarce and thus more challenging. In addition to the fisheye camera dataset mentioned in Table 9 in `Appendix A.2` where we can achieve a more obvious advantage over DualPose or SSPCM, we also intend to apply our findings to ego keypoint detection or hand keypoint detection. They are equally more important visual foundation capabilities to existing embodied intelligence research.

---

> ### Author Response · Authors · 2024-11-19
> **Official Responses by Authors**
>
> ***W3: Comparison with SSPCM***
>
> **R3:** As described in the setting S3 and shown in Table 5, except for SSPCM, our method still has an advantage over other fully supervised methods and the semi-supervised method PoseDual. We think there are two possible reasons: fewer training epochs and less network parameters. In fact, after submitting the paper, we trained another 400 epochs according to this setting (based on the backbone HRNet-w48, setting the batch size to 16 and using 8 A100s with 80GB of each), which took about 20 days. The final mAP and mAR values were 77.4% and 82.3%, respectively, which still did not exceed SSPCM (77.5% and 82.3%). We could not find any other better explanation, and can only conjecture that SSPCM uses a triple-network framework to bring better generalization on the COCO test-dev that has never been seen during training.
>
> This phenomenon is indeed very weird and incomprehensible, considering that our results in other settings are significantly better than SSPCM (please refer Tables 3, 4, 8 and 9). To this end, we tried to contact the authors of SSPCM to seek their final trained model weights based on HRNet-w48 for self-testing and comparison, but we never received a response so far. And their public training code does not contain the details of this part.
>
> ***
> ***W4: Performance on the test-set***
>
> **R4:** Similar to our explanation in W2, it is reasonable to achieve a small absolute performance improvement on COCO test-dev where the accuracy indicator is close to saturation when using the most powerful backbone HRNet-w48. In fact, the semi-supervised HPE methods shown in Table 5 performed almost the same, which makes it difficult to significantly and objectively distinguish the advantages and disadvantages of each method. Nevertheless, we present it here for complete comparison with previous methods. Considering that the main purpose of this paper is to use semi-supervised algorithms to try to improve network performance when labels are scarce, we believe that the setting based on smaller-scale labeled data can more sensitively reflect the advantages of one SSHPE method. Please refer results in Tables 3 and 9, which contain experiments under very small annotation rates.
>
> ***
> **W5: Instructions on dataset setup in S1**
>
> **R5:** Firstly, for the dataset setup in setting S1, we followed the previous work PoseDual in order to maintain fairness and reproducibility. Specifically, we select data from the first 1K, 5K, and 10K samples of the training dataset as labeled set, and the remaining samples are used as the unlabeled set. The randomness here just follows the data representation in PoseDual. To avoid ambiguity, we have corrected the statement in our main paper (marked in red color in Section 5.1). We are sorry that this oversight caused confusion to the reviewers.
>
> As additional supplementary information, when implementing the code, we can set the number of labeled samples as TRAIN_LEN in the configuration file, and directly select the first TRAIN_LEN samples when initially loading the training dataset. You can refer to the original PoseDual code to confirm this in https://github.com/xierc/Semi_Human_Pose/blob/master/lib/dataset/coco.py#L138
>
> Secondly, we totally agree that the conclusions drawn from testing and evaluation on small-scale data may not necessarily be generalized to other datasets. Therefore, we repeated the comparison in setting S1 and Table 3 by replacing the COCO dataset into MPII dataset. Specifically, we conducted experiments using the first 1K samples as labeled data and the left 39K samples as unlabeled data in MPII. The validation set of MPII is used to evaluate. The backbone is ResNet-18. The final comparison results are shown below.
>
> ```
> ---------------------------------------------------------------------------------
> Methods 	| Hea 	| Sho	| Elb	| Wri	| Hip	| Kne	| Ank	| Total
> ---------------------------------------------------------------------------------
> Supervised	| 89.6	| 84.8	| 72.0	| 58.4	| 57.8	| 49.4	| 41.2	| 65.3
> PoseCons	| 92.7	| 87.6	| 74.5	| 67.9	| 72.3	| 64.2	| 59.4	| 75.2
> PoseDual	| 93.3	| 88.4	| 75.0	| 67.3	| 72.6	| 65.3	| 59.7	| 75.6
> SSPCM		| 93.5	| 90.6	| 80.2	| 71.3	| 75.9	| 68.9	| 62.3	| 78.3
> Ours (Single)	| 94.1	| 91.1	| 80.5	| 72.2	| 76.3	| 69.2	| 62.8	| 79.1
> Ours (Dual)	| 94.7	| 92.4	| 81.2	| 73.3	| 76.8	| 70.6	| 63.9	| 79.7
> ---------------------------------------------------------------------------------
> ```
>
> Not surprisingly, our method still maintains a clear lead in performance, both in terms of overall accuracy and the specific accuracy of each joint. These experiments once again demonstrate that our method is indeed universally effective and superior across different datasets. We have updated  these results in our revised paper in **Appendix A.2**.
>
> ***
> **Continue in the next comment.**

---

> ### Author Response · Authors · 2024-11-19
> **Official Responses by Authors**
>
> ***W6: Explanation of results in Table 8***
>
> **R6:** Similar to our explanation in W2 and W4, the small absolute performance improvement on the MPII test-set is completely understandable when using the powerful backbone HRNet-w32. In fact, the MPII dataset was released earlier than the COCO dataset, so it faces a more severe problem of accuracy saturation, or cannot keenly reflect the differences between SOTA algorithms. As an alternative, we suggest that reviewers can refer to the Table 11 in **Appendix A.2**, which shows obvious advantages of our method over the other two SOTA algorithms (PoseDual and SSPCM) when the label rate is lower.
>
> ***
> **Q1: Division criteria in Table 5**
>
> **A1:** In fact, the upper, middle and lower areas in Table 5 show CNN-based fully supervised HPE methods, transformers-based fully supervised HPE methods and semi-supervised HPE methods respectively. We apologize for not stating this clearly, and we have included the corresponding statement in the revised paper (marked in red color in the description of setup S3).
>
> ***
> **Q2: Training efficiency of our method**
>
> **A2:** This is a good and important question. In order to fairly and reasonably reflect the efficiency of our method, we follow the setting S1 (using ResNet-18 as the backbone, batch size is set to 32, total training epochs are 30, and the amount of labeled data is 1K), and conduct each experiment on four 3090 graphics cards (with each containing 24 GB memory) to compare the training time of our method with that of PoseCons and PoseDual. The strong augmentation used by PoseCons or PoseDual is $T_{JC}$. Considering that our method often uses different strong augmentations, their computation is not the main bottleneck. Therefore, in order to be fair, all strong augmentations in our method are also replaced into $T_{JC}$. Assuming that the total training time of PoseCons is one unit time T0, which is actually about 7 hours. Then the total training time of running other methods is summarized as follows.
>
> ```
> -----------------------------------------------------------------------------------------------------
> Method	| PoseCons	| Ours(Single,2#)	| Ours(Single,3#)	| Ours(Single,4#)	|
> -----------------------------------------------------------------------------------------------------
> Time	|  T0		| 1.36 * T0		| 1.50 * T0		| 1.83 * T0		|
> -----------------------------------------------------------------------------------------------------
> Method	| PoseDual	| Ours(Dual,2#)		| Ours(Dual,3#)		| Ours(Dual,4#)		|
> -----------------------------------------------------------------------------------------------------
> Time	| 2.49 * T0	| 2.62 * T0		| 2.88 * T0		| 3.14 * T0		|
> -----------------------------------------------------------------------------------------------------
> ```
>
> where an integer with the marker # in our method means how many multi-path losses are used. From these results, we can see that when using four-path losses, although the training time increases, it is still faster than PoseDual (1.83\*T0 vs. 2.49\*T0). Referring to the quantitative results in Table 3 of the main paper, our method based on single-network using four-path losses achieves higher mAP than PoseDual. In addition, when using dual networks with four-path losses, the total training time does not increase significantly (2.49\*T0 vs. 3.14\*T0). These indicate that our method is both efficient and effective. We have added these analyses in **Appendix A.5**.
>
> ***
> ***Q3: The significance of Figure 7***
>
> **A3:** Although Table 3 has shown the quantitative comparison results, we still cannot intuitively and quickly perceive the performance differences between different methods under different annotation rates from these plain numbers. Therefore, we considerately visualize these mAP values in Figure 7. From it, we can easily see that our method not only obtains leading results, but also achieves a more obvious leading advantage when the data annotations are scarcer. This property is the most important and valuable characteristic of semi-supervised algorithms.
>
> ***
> Your feedback has been invaluable in helping us improve the clarity and comprehensiveness of our research. We are committed to addressing these points and enhancing the overall quality of our paper.

---

### Official Review · Reviewer_iqhT · 2024-10-31

**Soundness:** 2
**Presentation:** 2
**Contribution:** 1
**Rating:** 3
**Confidence:** 4

**Summary:**

The authors propose a method to enhance semi-supervised human pose estimation (SSHPE) through synergistic data augmentation and multi-path consistency training. Instead of creating isolated or complex augmentations, they combine existing augmentations in a complementary way that intuitively benefits SSHPE, achieving stronger results through these collaborative transformations. For consistency training, they forgo traditional multi-network stacking in favor of training a single network with multi-path consistency losses across multiple augmented views of the same unlabeled image batch, yielding both efficiency and accuracy gains.

**Strengths:**

1.	Focus on a Practical Problem: The paper addresses a fundamental and labor-intensive challenge in 2D human pose estimation (HPE)—the need for extensive labeled data. By leveraging semi-supervised learning (SSL) to utilize unlabeled data, the approach targets a practical solution that could reduce the dependency on costly and time-consuming data annotation.
2.	Innovative Augmentation Strategies: The authors identify a unique contribution by combining existing data augmentations to create “easy-hard” augmentation pairs that introduce a wider difficulty spectrum. This approach leverages the synergistic effects of established augmentations to generate novel HPE-specific augmentations, potentially enhancing model robustness by training on more challenging, noise-introduced variations of data.
3.	Simplified and Effective Consistency Training: Instead of relying on complex multi-network architectures, the paper proposes a single-network design that optimizes multiple losses through sequential multi-path predictions on heavily augmented data. This simplified approach is both interpretable and efficient, making it accessible for broader implementation while maintaining strong performance gains.
4.	Commitment to Open Science: By releasing code for academic use, the authors support transparency and reproducibility, enabling other researchers to build upon this work and further validate the approach across diverse HPE tasks and datasets.

**Weaknesses:**

### Limited Novelty

The proposed method in this paper lacks sufficient novelty and fails to contribute meaningful advancements to the field of semi-supervised human pose estimation (SSHPE). Human pose estimation, as noted, is already a thoroughly researched area with many well-established methods for both data augmentation and consistency training. Specifically, the augmentations Joint Cutout (TJC) and Joint Cut-Occlude—are already known [2]. Similarly, the concept of consistency training has been explored extensively in prior work (e.g., Xie et al., 2021; Moskvyak et al., 2021; Li & Lee, 2023; Huang et al., 2023), reducing the originality of the current approach.

Additionally, this paper’s methodology appears closely aligned with SSPCM [2], sharing a similar approach to consistency training without offering meaningful differentiation both in terms of novelty and accuracy (Table 3,4). The methods do not extend beyond existing approaches like POST [1], which employs consistency training alongside augmentation but tackles a more complex problem involving significant distribution shifts—something that requires a more robust approach. Given that the paper neither introduces novel methodologies nor demonstrates improvements over current state-of-the-art techniques, its contributions are limited in both theoretical and practical impact. Consequently, the paper falls short in providing sufficient novelty or relevance to warrant further consideration.

The current approach to selecting augmentations is somewhat rudimentary; exploring more advanced augmentation generation paradigms, such as PoseAug [3], could substantially enhance the contribution by introducing more targeted, pose-specific transformations. To further strengthen the paper’s impact, the authors could also address more complex challenges within semi-supervised human pose estimation, such as managing significant distribution shifts or effectively handling occlusions, both of which would demonstrate the robustness and adaptability of the proposed method in diverse and realistic scenarios.

### Lack of experminets

The proposed method for semi-supervised human pose estimation (SSHPE) would benefit from a deeper comparison with existing foundation models, particularly in a zero-shot setting. For instance, models like Sapiens [4] have demonstrated strong generalization in zero-shot applications, and evaluating the proposed SSHPE approach against such foundation models would clarify its relative strengths and weaknesses. Additionally, the paper overlooks a critical experiment involving the use of high-accuracy 3D pose estimation models, such as those utilizing the SMPL model, which effectively parameterizes human poses, handles occlusions, and supports 3D-to-2D projection via camera transformations. Testing whether training a 2D pose estimation model under a semi-supervised setting indeed outperforms simply projecting a state-of-the-art 3D model (e.g., BEDLAM [5], which is trained solely on synthetic data) into 2D, would be crucial in establishing the necessity of the SSHPE approach. Given BEDLAM’s potential for zero-shot 2D projection, it is likely that both Sapiens and BEDLAM could outperform the proposed SSHPE model without additional training.

These omissions raise an essential question about the need for SSHPE approaches in the presence of robust, generalizable foundation models and accurate 3D pose estimators. To justify the SSHPE setting, it is recommended that the authors add a dedicated section explaining its relevance and necessity. Additionally, expanded experimental results are needed to establish a clear advantage of the proposed method over these established models, particularly to demonstrate that the proposed approach fills a specific gap that existing foundation models and 3D estimators do not.

### Results

Loss Function Effectiveness: Currently, the paper does not provide sufficient insight into the effectiveness of the proposed loss functions,  $\mathcal{L}_u$  and  $\mathcal{L}_S$. Including an ablation study to examine the individual impact of these losses on labeled and unlabeled datasets would provide a clearer understanding of their contributions. This would help substantiate the effectiveness of each component and offer more transparency on how these loss functions drive model performance.

Comparison with UDA Methods: The semi-supervised setting described shares similarities with unsupervised domain adaptation (UDA) approaches, suggesting a relevant opportunity for comparison. Including benchmark results against state-of-the-art UDA methods, such as UDAPE [6], which leverages consistency loss minimization, would highlight the unique value of the proposed method. Adding these comparisons in Table 4 would show how the approach fares relative to established UDA methods and underscore its novelty and advantage in this space.

### Lack of clarity
The paper lacks essential clarity, making it challenging for readers unfamiliar with the field to follow the methodology and results. Specifically:

Undefined Variables: Key variables such as  $u$, $l$, $N$, and $M$ in lines 181-182 are not clearly defined, which disrupts the reader’s ability to interpret the equations and understand their significance within the proposed method.

Unclear Justification for Downsampled Heatmaps: In line 161, there is a reference to predicting downsampled heatmaps without any clear explanation or citation to support this choice. Given that downsampling can impact resolution and accuracy, it’s essential to provide references to prior methods that utilize this approach and clarify why it’s suitable in this context.

### References

[1] Prior-guided Source-free Domain Adaptation for Human Pose Estimation, ICCV 2023
[2] Semi-Supervised 2D Human Pose Estimation Driven by Position Inconsistency Pseudo Label Correction Module, CVPR 2023
[3] PoseAug: A Differentiable Pose Augmentation Framework for 3D Human Pose Estimation, CVPR 2021
[4] Sapiens: Foundation for Human Vision Models, ECCV 2024
[5] BEDLAM: A Synthetic Dataset of Bodies Exhibiting Detailed Lifelike Animated Motion, CVPR 2023
[6] A Unified Framework for Domain Adaptive Pose Estimation, ECCV 2022

**Questions:**

It would be helpful if the authors could address key areas including Novelty and Differentiation, Comparison with Foundation Models, Justification of the SSHPE Setting, as well as provide insights on the suggested experimentation, ablation studies, and clarity improvements. Clarifying these aspects would significantly enhance the understanding of the paper’s contributions. Addressing these points would also help position the work within the broader landscape of current methods, highlighting any distinct impact and practical relevance.

# UPDATE (After Discussion Period)

I appreciate the author's response and thanks for sharing a detailed response and addressing concerns related to novelty, experimentation and ablation studies. The statement made by the authors, "Almost all of the above methods have not been or cannot be tested on the 2D HPE datasets used in this paper," appears unjustified and lacks merit. Several existing approaches, including foundational models like SAPIENS and BEDLAM, offer the capability to perform zero-shot evaluations on 2D HPE datasets without requiring additional training. Conducting such evaluations is straightforward and would provide a meaningful benchmark against proposed methods. This comparison is critical to establish the relevance and competitive performance of the proposed method in the broader context of foundation models.

Without such comparisons, the study's scope of semi-supervised learning remains speculative, as it lacks a clear understanding of how the proposed method measures up to existing works. Hence, **I would like to retain my current rating**, as the lack of rigorous comparative evaluation constitutes a significant shortcoming.

---

> ### Author Response · Authors · 2024-11-19
> **Official Responses by Authors**
>
> Dear Reviewer iqhT,
>
> Thank you for your detailed review, positive appreciation and critical points about our paper. Your feedback is essential for refining our work, and we have addressed each of your concerns below.
>
> ***
> ***W1: Concerns about novelty***
>
> **R1:** Firstly, the research goal of this paper is to try to propose a new paradigm based on existing basic augmentations, so as to obtain a new advanced augmentation combination conveniently and efficiently. Therefore, we adopt the Joint Cutout augmentation in the existing similar work PoseDual and the Joint Cut-Occlude augmentation in SSPCM as elements of the basic augmentation set. These do not conflict with the innovation of this paper.
>
> Secondly, our proposed superior augmentation generation paradigm and multi-path consistency loss strategy do not overlap with the methods SSPCM or POST. Specifically, SSPCM is a triple-network framework that improves PoseDual by adding an auxiliary network to provide additional help for examining the quality of pseudo-labels. Our approach is different from SSPCM in both structure and core strategy. And our quantitative results in Tables 3 and 4 are clearly better than those of SSPCM. While, POST applies the most commonly used mean-teacher architecture in unsupervised domain adaptation to gradually update the teacher model to better predict pseudo labels. When training, only the student model uses backward propagation to update parameters, and the teacher model uses exponential moving average (EMA) to update parameters without gradients. It can be seen that POST is completely different from the teacher-student alternating network used in our paper.
>
> Finally, our proposed strong augmentation generation paradigm is experimentally demonstrated to be concise and effective for the task of 2D HPE. While, PoseAug is for solving the 3D HPE problem, which usually desires a 2D-to-3D pose estimator/lifter, whose input is a series of detected 2D keypoints. It is a completely different field from our 2D keypoint detection using the RGB image as input. This means that augmentations used by these two fields are not mutually compatible.
>
> ***
> ***W2: Lack of experiments***
>
> **R2:** We appreciate the reviewer pointing out many other topics and researches that are somehow related to our study, such as 3D human pose estimation [3], 3D human pose and shape estimation [5], source-free domain adaptive human pose estimation [1], and human vision foundation model [4]. Although these methods show various advantages and generalization in human-related topics, they are not completely consistent with the research content SSHPE in this paper. Almost all of the above methods have not been or cannot be trained and tested on the 2D HPE datasets COCO and MPII used in this paper. Therefore, we cannot fairly or quickly compare our method quantitatively with these cross-proposition methods. Nevertheless, we agree that in future research, we would try to obtain the 2D human keypoint prediction results of the large foundation model or 3D human model in a zero-shot manner in order to compare with our method. This is indeed a more interesting and promising research area.
>
> ***
> ***W3: Provide more results***
>
> **R3:** Regarding the loss function effectiveness, we agree that the weight of the supervised loss and the unsupervised loss is a very important hyper-parameter. In the experiments of this paper, in order to compare fairly with previous methods (including PoseDual and SSPCM), we follow the same setting as them and use the same weight for these two losses. We did not deliberately optimize this hyper-parameter, which would be unnecessary or unfair.
>
> Regarding the unsupervised domain adaptation HPE methods, we agree that it has something in common with the topic SSHPE in this paper, but these two research fields cannot be confused. The UDA HPE method is only applicable when there is a clear domain difference between the labeled source domain and the unlabeled target domain. However, our semi-supervised HPE research does not emphasize the domain difference between the labeled set and the unlabeled set. Therefore, it is unfair and inappropriate to directly compare the SOTA algorithm UDAPE [6] with SSHPE approaches including PoseDual, SSPCM and our method in Table 4. Nevertheless, we expect that our future research will apply the core innovations proposed in this paper (including the new superior augmentation and multi-path consistency loss) to address the UDA HEP problem.
>
> ***
> ***W4: Improve content clarity***
>
> **R4:** Thank you for pointing out these omissions. We have clarified these details in our updated paper. Please refer to the first paragraph of Section 3, where the revised part is marked in red color.
>
> ***
> We hope these revisions address your concerns and provide a more comprehensive understanding of our method and its unique contributions. We are dedicated to making the necessary updates to enhance the clarity and impact of our paper.

---

### Official Review · Reviewer_4hx9 · 2024-11-03

**Soundness:** 2
**Presentation:** 3
**Contribution:** 2
**Rating:** 6
**Confidence:** 4

**Summary:**

This paper aims to boost semi-supervised human pose estimation from two perspectives: data augmentation and consistency training. For data augmentation, this paper conducts empirical studies to get the easy-hard rank of augmentations (joint cutout, joint cut-occlude, randaugment, cutmix, etc.) for human pose and recommend two augmentation combinations (i.e., a cutout after joint cut-occlude and a cutmix after joint cutout). For consistency training, the paper proposes using one easy augmentation to supervise multiply hard augmentation. Experiments on COCO and MPII datasets show that the proposed method outperforms existing SOTA methods.

**Strengths:**

- Based on the results in Table 3, the proposed method exceeds previous SOTA results.
- The rank and the combination of augmentations for semi supervised learning is important, and this work first try to dive deeper into this study.

**Weaknesses:**

1. L.95 introduces a principle "Do not combine MixUp-related augmentations" while L.224 P1 claims that "A global $T_{MU}$ does not make sense for the HPE task." It would be better to provide more details. What is the meaning of global? Moreover, it seems mixup is used in some heatmap-based works like [R1], and it works well.

- [R1] From Synthetic to Real: Unsupervised Domain Adaptation for Animal Pose Estimation. CVPR2021.

2. It would be better to include $T_{CM}$, $T_{CO}$ and $T_{MU}$  in Table 1.

3. Section 3.1 concludes that the rank and the combination of data augmentations are entirely based on empirical studies.  For example, it is easy to understand $T_{A60}$ is harder than $T_{A30}$, but it is not intuitive to understand $T_{JC}$ is harder than $T_{CO}$. I am wondering if any insights or theoretical analysis can be provided to help readers understand more intuitively. For example, L. 228 claims that "we thus nominate the most likely superior combinations: TJOCO and TJCCM". The two combinations are useful. However, can we have any insights or principles for other augmentations which not discussed in this paper. It would be meaningful if we can quickly rank any new augmentations.

4.  Section 3.1 did not discusses the parameters of each augmentation. I am not sure the default parameters of each augmentation. Moreover, I am wondering if the parameters will have significant influences on the rank and the combination. For example, Dual-Network (Xie et al., 2021) prefers to use JC 5 and RA 20 to show the parameters directly.

5. Section 5.3 discusses more augmentations and training techniques. But they are not ablation study. It would be better to clearly show the baseline and the effect of each module/strategy.

6. It would be better to show some qualitative results in the main paper or in the supplementary.

7. I am not sure the comparison between the multi-path consistency loss and others is fair enough due to the difference of batch size. It would be better to compare the pair {$I_{e1}$,...,$I_{en}$} + {$I_{h1}$,...,$I_{hn}$} with {$I_{e}$} + {$I_{h1}$,...,$I_{hn}$}, and also discuss the influence of batch size of other SOTA methods.

8. Minor comments
- there is an incorrect usage of cite in L.47.
- there are incorrect punctuation in formulas like Eqs.3 and 4.

**Questions:**

See weaknesses.

**Details Of Ethics Concerns:**

No.

---

> ### Author Response · Authors · 2024-11-19
> **Official Responses by Authors**
>
> Dear Reviewer 4hx9,
>
> We greatly appreciate your thorough review, constructive feedback and praise for groundbreaking on our paper. Your points have helped us identify areas for clarification and improvement. Please find our responses to your queries below.
>
> ***
>
> ***W1: Why not using MixUp?***
>
> **R1:** First, the MixUp operation is to superimpose and mix two images of the same size pixel by pixel, and the global mixing ratio $\alpha$ is randomly generated in advance. We summarize this process as ${Img} = \alpha*{Img}_1 + (1-\alpha)*{Img}_2$. Global here means the entire image is superimposed, using a predetermined blending ratio value.
>
> In addition, MixUp is indeed used in the heatmap-based method like [R1] and performs well. We think that the main emphasis is on using MixUp to alleviate the problem of domain discrepancies in the domain transfer process of sim2real. And making it as difficult as possible for the model to distinguish between the source domain and the target domain is a common practice in the unsupervised domain adaptation from synthetic to real field. Therefore, it is reasonable that MixUp may be helpful in [R1] after mixing synthetic images and real images as input. However, our paper does not emphasize the domain adaptation setting. In each independent experiment, all labeled and unlabeled images used are from the real world and there is no obvious inter-domain differences. At this point, we lack a theoretical basis for using the MixUp operation, and our related experiments in Figure 2 and Table 1 also verify that MixUp is inappropriate for solving the SSHPE problem.
>
> ***
>
> ***W2: Modification of Table 1***
>
> **R2:** Thank you for your suggestion. We have added the data of these three base augmentations in Table 1 to facilitate quick comparison. At the same time, the corresponding convergence curves in Figure 4 have also been updated.
>
> ***
>
> ***W3: Principles for combining augmentations***
>
> **R3:** First, we need to rank the difficulty of some well-known basic augmentations, which we have already shown in Figure 2 and the first paragraph of Section 3.1. We also provide a theoretical analysis of the superior augmentation in **Appendix A.1** as an explanation support. Next, we need to recognize the synergy between basic augmentations, that is, which and how many base augmentations to combine have general rules to follow. We provide an intuitive explanation in the second paragraph of Section 3.1, and its quantitative verification is shown in Table 1 and Figure 4. In practice, when combining different basic augmentations, we summarized three simple operating principles by integrating the above two pieces of information. These are introduced in the third paragraph of Section 3.1, and there are sufficient empirical experiments in the fourth paragraph to support the feasibility of these principles. These experiences guide us to quickly find the optimal augmentation combination instead of experimenting with all possible combinations one by one. It should be noted that the empirical researches involved here are not the essential reason for deducing optimal combinations, but for quantitative verification.
>
> For $T_{JC}$ and $T_{CO}$, namely Joint Cutout and trivial Cutout, they both generate several small square zero-value masks/patches in the image. The difference is that the former mainly generates occlusion around the keypoint area of the human body, while the position of the latter is random. Obviously, the former will bring greater challenges to the keypoint prediction task because of more frequent occlusion. The same explanation applies to $T_{JO}$ and $T_{CM}$, except that these patches of them are cropped from another image instead of using zero values.
>
> Finally, for a newly added augmentation, we only need to repeat the first step, that is, to determine its difficulty ranking among basic augmentations. Then we can get a better augmentation combination according to the three principles proposed. For example, we have verified in Table 6 that the augmentation YOCO can produce synergistic effects with RandAugment or TrivialAugment. Then we can rank it, and recommend new and better augmentation combinations, such as $T_{JOCO+YOCO}$ and $T_{JCCM+YOCO}$. These are essentially similar to the procedures we have demonstrated when selecting out $T_{JOCO}$ and $T_{JCCM}$, so the experiments are not repeated in our main paper due to space limitations and minor significance.
>
> ***
>
> **Continue in the next comment.**

---

> ### Author Response · Authors · 2024-11-19
> **Official Responses by Authors**
>
> ***W4: Parameters of each augmentation***
>
> **R4:** Thanks a lot for pointing out this important detail. The hyper-parameters involved in each augmentation are indeed important. In order to make a fair comparison, each basic augmentation we selected is derived from various compared methods without additional fine-tuning. For example, the parameters of Joint Cutout are the same as those in PoseDual which used JC5, and the parameters of Joint Cut-Occlude are the same as those in SSPCM which used JO2. We list these parameters in **Appendix A.4** so that readers can quickly and clearly know these details.
>
> ***
> ***W5: Clearer ablation studies***
>
> **R5:** Thank you for raising this concern. In fact, when we were conceiving the overall framework of this paper, we also struggled with the issue of how to present the ablation experiments. After much deliberation, we finally decided to present the ablation results of our proposed augmentation combination and multi-path loss in advance in Sections 3.1 and 3.2 as quantitative data for empirical studies, so that readers can quickly understand and get into the two major themes, data augmentation and consistency training. Therefore, the results in Tables 1 and 2 are essentially ablation studies. To some extent, the results in Table 3 also have the same effect, showing the comparison results of different network structures and label annotation rates. In Section 5.3, we present and compare additional important components, including other similar augmentation combinations and different training techniques. We hope that this arrangement can satisfy most readers and help them capture the key points.
>
> ***
> ***W6: Showing qualitative results***
>
> **R6:** Thank you for your sincere suggestion. We have added qualitative visualization comparison results in **Appendix A.3**, mainly including the conventional human images from the COCO val-set and the fisheye camera images from the WEPDTOF-Pose dataset. We expect these results will make our advantages more intuitively demonstrated.
>
> ***
> ***W7: The multi-path consistency loss***
>
> **R7:** This is indeed a good question. In fact, when designing the ablation experiments with PoseCons using a single-path loss, we chose a fixed batch size 32 to perform all experiments (all using backbone ResNet-18). All relevant results can be found in Table 1. When using multi-path consistency losses, we still set the batch size to 32, including the two-path losses (including $m_{1} \sim m_{7}$ and $m_{9} \sim m_{11}$) and four-path losses (including $m_{8}$ and $m_{12}$) in Table 2. Therefore, in final comparative experiments (see Table 3), we still keep the batch size as 32 and use the optimal four-path losses. Now, in order to investigate the possible impact of different batch sizes, we report the effects of PoseCons and PoseDual when the batch size is 128.
>
> ```
> -----------------------------------------------------------
> Method		| Nets.	| Losses | BS   | 1K   | 5K   | 10K
> -----------------------------------------------------------
> PoseCons	| 1	| 1	 | 32   | 42.1 | 52.3 | 57.3
> PoseCons	| 1	| 1	 | 128  | 42.3 | 52.6 | 57.5
> PoseDual	| 2	| 1	 | 32   | 44.6 | 55.6 | 59.6
> PoseDual	| 2	| 1	 | 128  | 44.9 | 58.7 | 59.6
> Ours (Single)	| 1	| 4	 | 32   | 45.5 | 56.2 | 59.9
> Ours (Dual)	| 2	| 4	 | 32   | 49.7 | 58.8 | 61.8
> -----------------------------------------------------------
> ```
>
> As can be seen, after increasing the batch size of PoseCons or PoseDual accordingly, the final mAP results under different labeling rates (e.g., 1K, 5K and 10K) did not get significantly better. This indicates that batch size does not have a large impact on the performance of existing methods.
> We also conducted additional experiments to address another concern, namely whether to use a single constant easy augmentation as input for multi-path losses (the pair {$I_e$} + {$I_{h_1},...,I_{h_n}$}, termed as 1-vs-n) or to use different easy augmentations multiple times as input (the pair {$I_{e_1}$,...,$I_{e_n}$} + {$I_{h_1}$,...,$I_{h_n}$}, termed as n-vs-n).
>
> ```
> ----------------------------------------------------------------------
> Method		| Nets.	| Losses | BS   | Input	 | 1K   | 5K   | 10K
> ----------------------------------------------------------------------
> Ours (Single)	| 1	| 4	 | 32   | 1-vs-n | 45.5 | 56.2 | 59.9
> Ours (Single)	| 1	| 4	 | 32   | n-vs-n | 45.6 | 56.4 | 59.8
> Ours (Dual)	| 2	| 4	 | 32   | 1-vs-n | 49.7 | 58.8 | 61.8
> Ours (Dual)	| 2	| 4	 | 32   | n-vs-n | 49.7 | 58.9 | 61.9
> ----------------------------------------------------------------------
> ```
>
> As shown in the table above, whether using 1-v-n augmented input or n-vs-n augmented input, the final mAP results obtained under various labeling rates are not significantly different. This is mainly because the used easy augmentation is always fixed (e.g., $T_{A30}$), so the input does not change in essence. We have added these additional ablation studies in **Appendix A.5**.
>
> ***
>
> **Continue in the next comment.**

---

> ### Author Response · Authors · 2024-11-19
> **Official Responses by Authors**
>
> ***W8: Revision of minor comments***
>
> **R8:** Thank you for pointing out these detailed typos. We have corrected them in the main text. At the same time, we have also added commas or periods at the end of other formulas to make the paper more standardized.
>
> ***
>
> Your feedback has been instrumental in enhancing the clarity and accuracy of our work. We are committed to making the necessary revisions to address these points thoroughly.

---

> > ### Comment · Reviewer_4hx9 · 2024-11-27
> >
> > I appreciate empirical studies and greatly value the additional experiments included in the response. Some issues have been resolved. However, based on other review and my own understanding, I feel that while I could raise my score, the paper still requires reorganization and further polishing. Especially,
> >
> > - For mixup, domain gaps do exist in the real world (e.g., daytime vs. nighttime, sunny vs. rainy), even these gaps may be smaller than those in syn-to-real situations.
> >
> > - For basic augmentations including rankings, combinations and parameter selections, due to so many variations, simply providing specific examples in this paper is not sufficient and convincing. General tools and guidelines might be more important, and should be added and highlighted in the very beginning of the main paper.

---

> > > ### Author Response · Authors · 2024-11-27
> > > **Official Responses by Authors**
> > >
> > > Thank you for your kind reply and heartfelt affirmation.
> > >
> > > For MixUp, it is indeed a widely used strategy to alleviate the inter-domain differences. However, we did not address the inter-domain adaptation problem in this paper, so we did not focus on it. We will actively explore its great potential in future work.
> > >
> > > As for the hyper-parameters of basic augmentations, this is indeed a very important component, almost the core of any kinf of data augmentation. Unfortunately, we need to make a fair and reliable comparison with previous similar works, so we did not explore it in depth. Perhaps our future work can focus on the study of AutoAug algorithms suitable for SSHPE tasks.
> > >
> > > Thank you again for your hard work in reviewing this paper.

---

### Official Review · Reviewer_TvMD · 2024-11-05

**Soundness:** 2
**Presentation:** 3
**Contribution:** 2
**Rating:** 5
**Confidence:** 5

**Summary:**

This paper presents a method for SSL 2D Human Pose estimation that improves existing methods from two aspects: data augmentation and better consistency loss design.

**Strengths:**

1. The paper is well-written and easy to follow
2. Personally, the idea of ranking basic augmentation looks interesting to me. From my own experience, heatmap-based prediction has different behavior from augmentations from traditional classifier-based models so this line of work looks interesting to me.
3. The experiment results show decent amount of improvement.

**Weaknesses:**

1. It seems that the dual network design is still a dominant approach based on experiment results. The single network with proposed components still cannot outperform dual network approaches.
2. How does the performance look like with ResNet-50? Some other work also evaluates with this model. Honestly, I don't really think ResNet-18 is that important these days.
3. The improvement on the coco training + coco wild seems to be marginal.

**Questions:**

See above

---

> ### Author Response · Authors · 2024-11-19
> **Official Responses by Authors**
>
> Dear Reviewer TvMD,
>
> Thank you for your high recognition, valuable feedback and insightful comments on our paper. We appreciate the opportunity to clarify and address your concerns. All our responses are as follows.
>
> ***
>
> ***W1: Effectiveness under a single network***
>
> **R1:** In this paper, we mainly use single-network to demonstrate the efficiency and effectiveness of the proposed optimal augmentations combination and multi-path loss strategies. In the experimental comparison, the results involving a single network are mainly concentrated in Tables 3, 4, and 9. When there is only a single network, our method always outperforms the baseline method, namely PoseCons. Please compare line 4 vs. line 8 in Table 3, and line 2 vs. line 5 in Table 9. Even when the baseline method PoseDual uses dual networks, our single network based results are still much better. Please compare line 5 vs. line 8 in Table 3, lines 2/8/14 vs. lines 5/11/17 in Table 4 and line 3 vs. line 5 in Table 9. These advantages based on a single network clearly demonstrate the effectiveness of our core method design.
>
> When we upgrade the proposed method to a dual network structure, the results are naturally better, indicating that it has become the dominant approach. And compared with SSPCM using triple networks, our dual-network approach still has obvious advantages in most cases, while the single-network approach also shows comparable performance.
>
> ***
>
> ***W2: Performance with using ResNet-50***
>
> **R2:** We have shown the comparison results based on ResNet-50 in Table 4, mainly including methods PoseDual, SSPCM and Pseudo-HMs. Actually, for a fair comparison, we follow the experimental settings in these compared methods and keep using ResNet-18 in Table 3 for convenient ablation researches, HRNet-w48 in Table 5 to compare performance limits, and HRNet-w32 in Tables 7 and 8 to highlight the best performance.
>
> In addition, we strongly agree that reporting the comparison results based on the more important ResNet-50 is more convincing than ResNet-18. Therefore, we replaced the backbone in Table 3 with ResNet-50 according to setting S1 and re-conducted the comparative experiments. The results are as follows.
>
> ```
> ------------------------------------------------------
> Method		| Nets.	| 1K    | 5K    | 10K   | All
> ------------------------------------------------------
> Supervised	| 1	| 34.8  | 50.6  | 56.4  | 70.9
> PoseCons	| 1	| 43.1  | 57.2  | 61.8  | ---
> PoseDual	| 2	| 48.2  | 61.1  | 65.0  | ---
> SSPCM		| 3	| 49.8  | 61.8  | 65.5  | ---
> Ours (Single)	| 1	| 49.3  | 61.4  | 65.2  | ---
> Ours (Dual)	| 2	| 51.7  | 62.9  | 66.3  | ---
> ------------------------------------------------------
> ```
>
> As shown in the table, similar to using ResNet-18, our method can still achieve a clear advantage. When using a single network, our method outperforms PoseCons and Posedual, while being comparable to SSPCM. And our dual-network based approach achieves significant advantages. We have updated our paper with these additional results in **Appendix A.2**.
>
> ***
>
> ***W3: Marginal improvement on COCO***
>
> **R3:** In our paper, the settings S2 and S3 are based on labeled COCO train-set and unlabeled COCO wild-set. For setting S2, we validate models in the COCO val-set, and report results in Table 4. Our approach has achieved not-insignificant advantages. Of course, the performance improvement gradually decreases as the number of parameters and sophistication of the backbones used grows (e.g., from ResNet-50 to ResNet-101 and HRNet-w48). This phenomenon is also true for previous compared methods such as PoseCons, PoseDual, SSPCM and Pseudo-HMs.
>
> For setting S3, we validate models in the COCO test-dev, and report results in Table 5. The same explanation applies to this similar phenomena. It should be noted that some SOTA supervised learning methods (e.g., UDP and ViTPose) in Table 5 also achieved similar performance, which indicates that the accuracy on this dataset is suspected to be saturated. These situations actually prevent us from objectively evaluating different methods on it.
>
> As an alternative, we recommend evaluating the pros and cons of different SSL methods by looking at the significant performance differences on human images with low annotation rates (see Table 3) or captured using unconventional fisheye cameras (see Table 9).
>
> For referring similar concerns, please refer to our responses to W2/W3/W6 raised by `Reviewer r3Sz`.
>
> ***
>
> We hope these clarifications address your concerns and contribute to a better understanding of our work. We are committed to making the necessary revisions in our paper to reflect these clarifications and enhance its overall quality.

---

### Author Response · Authors · 2024-11-19
**Submission of Revised Paper**

We are writing to inform you that we have submitted the revised version of our paper. We would like to express our sincere gratitude for your insightful and constructive comments and feedback. Your expert critiques have been invaluable in guiding the improvements we have made to our paper.

In response to the points raised during the review process, we have made the following comprehensive revisions:

* **Clearer Description:** We have added explanations of some variables and references of default input processing in the problem definition of Section 3 (for ***W4*** by `Reviewer iqhT`). We also reiterated that the labeled datasets of size 1K, 5K, and 10K from the COCO training set are selected in a fixed pattern and can be reproduced in Section 5.1 (for ***W5*** by `Reviewer r3Sz`). And we explained the rationale for the regional divisions in Table 5 (for ***Q1*** by `Reviewer r3Sz`).

* **More Performance Comparison Details:** In Appendix A.2, we added new comparison experiments based on ResNet-50 in Table 10 (for ***W2*** by `Reviewer TvMD`) and new comparison experiments using MPII dataset in Table 11 (for ***W5*** by `Reviewer r3Sz`) as more convincing supplements to Table 3 based on ResNet-18 and COCO dataset in the main paper.

* **Qualitative Visualization Comparison:** In Appendix A.3, we added the visualization  comparison of predicted results on COCO val-set and WEPDTOF-Pose test-set of various methods (for ***W6*** by `Reviewer 4hx9`).

* **Parameters of Basic Augmentations:** In Appendix A.4, we described in detail the hyper-parameters of basic augmentations used in this paper (for ***W4*** by `Reviewer 4hx9`).

* **Additional Ablation Studies:** In Appendix A.5, we tested and analyzed the impact of different batch sizes or easy augmented input ways on the multi-path loss function in Tables 13 and 14 (for ***W7*** by `Reviewer 4hx9`). We also analyzed and verified the efficiency and effectiveness of multi-path consistency loss training in Table 15 (for ***Q2*** by `Reviewer r3Sz`).

We believe that these comprehensive revisions have notably elevated the quality, clarity, and robustness of our research. We are hopeful that the paper now aligns more closely with the esteemed standards of the conference. We are grateful for the opportunity to refine our work and appreciate the dedication and effort you have invested in reviewing our paper.

Thank you once again for your invaluable assistance in enhancing the quality of our work. We look forward to your continued guidance and are hopeful for favorable consideration of our paper.

---

### Author Response · Authors · 2024-12-01
**More New Experiments on Semi-Supervised Human Hand Keypoints**

Dear Area Chairs and Reviewers,

Recently, in order to further verify the wide applicability and superiority of our proposed method MultiAugs, we have conducted comparative experiments on the human hand keypoint detection task which is very similar to the SSHPE task for the human body. The final conclusion is still impressive. These experimental results can serve as a strong supplement to the responses to `reviewer TvMD` in **W3R3**, `reviewer iqhT` in **W3R3**, `reviewer r3Sz` in **W3R3** and **W4R4**, and `reviewer z46M` in **W3R3**. Please consider it as an additional reference.

Specifically, we selected relevant human hand images and corresponding keypoint annotations from the COCO-WholeBody dataset [R1] as the annotated dataset (where the number of hand keypoints is 21), and obtained a train-set and a val-set containing approximately 76K and 3.8K samples, respectively. In addition, we used the BPJDet detector [R2] to extract approximately 118K samples from COCO wild-set as the unlabeled dataset. Then, we repeated setups **S1** and **S2** in the experimental setting and obtained results similar to those in Tables 3 and 4, respectively. The specific experiments and conclusions which are completely new are as follows.

***
```
------------------------------------------------------
Method		| Nets.	| 1K    | 5K    | 10K
------------------------------------------------------
Supervised	| 1	| 33.4  | 38.9  | 41.9
PoseCons	| 1	| 52.1  | 57.4  | 59.4
PoseDual	| 2	| 55.9  | 60.1  | 62.0
SSPCM		| 3	| 58.8  | 62.4  | 64.5
Ours (Single)	| 1	| 56.3  | 60.9  | 64.1
Ours (Dual)	| 2	| 62.8  | 65.5  | 67.4
------------------------------------------------------
```
First, we still used ResNet18 as the backbone in the setup **S1**, and then conducted comparative experiments on the methods SimpleBaseline, PoseCons, PoseDual, SSPCM, and the proposed MutliAugs. As shown in the table above, our method is still significantly better than the PoseCons using a single network or the PoseDual using two networks when using a single-network structure. When we use a dual-network structure, our method is significantly better than the SSPCM using a triple-network. These results and the trends they reveal are consistent with Tables 3, 9, 10, and 11 shown in our paper.

***
```
----------------------------------------------------------
Method		| Backbone	| Nets	| AP	| AR
----------------------------------------------------------
Supervised	| ResNet50	| 1	| 62.1	| 74.9
PoseDual	| ResNet50	| 2	| 65.9	| 78.3
SSPCM		| ResNet50	| 3	| 66.3	| 78.8
Ours (Single)	| ResNet50	| 1	| 66.8	| 79.2
Ours (Dual)	| ResNet50	| 2	| 67.4	| 79.7
----------------------------------------------------------
Supervised	| ResNet101	| 1	| 64.5	| 76.8
PoseDual	| ResNet101	| 2	| 68.0	| 80.2
SSPCM		| ResNet101	| 3	| 69.1	| 81.1
Ours (Single)	| ResNet101	| 1	| 69.9	| 81.7
Ours (Dual)	| ResNet101	| 2	| 71.8	| 83.3
-----------------------------------------------------------
```
Then, we followed the setup **S2**, and conducted experiments on COCO-WholeBody by using the annotated train-set as the labeled set and the COCO wild-set as the unlabeled set. Due to time constraints, we only performed various experiments under backbones ResNet50 and ResNet101 and did not have time to perform tests on HRNet-w48. The quantitative results on the val-set are summarized in the table above. From these results, we can find that our method still has an undoubted advantage, which is consistent with the phenomenon shown in Table 4.

The above experimental results show that the advanced augmentation combination and multi-path consistency loss strategy we proposed is indeed sustainable, effective and easy to promote. For example, for keypoint detection tasks, whether it is human body or hand, MultiAugs has reliable transferability and great potential in versatility. We expect that these experiments can further help demonstrate the core contribution of this paper.

Best Regards,

All Authors

***

**References**
* *[R1] Whole-Body Human Pose Estimation in the Wild, ECCV2020*
* *[R2] BPJDet: Extended Object Representation for Generic Body-Part Joint Detection, TPAMI2024*

---

### Meta-Review · Area_Chair_ms24 · 2024-12-17

**Metareview:**

This paper received five reviews, and the authors submitted a response addressing the queries raised. While two reviewer shifted to a positive stance after the rebuttal, albeit with some reservations, the others retained their initial ratings, which were largely below the acceptance threshold. Key concerns included the paper's marginal improvements over existing work and the lack of essential experiments for fair comparisons.

The overall consensus is that the paper requires significant revisions to meet publication standards. After careful consideration, the Area Chair panel decided not to accept the paper in its current form. We encourage the authors to address the reviewers' feedback thoroughly and submit a stronger, more comprehensive version in the future.

**Additional Comments On Reviewer Discussion:**

Post rebuttal reviewers 4hx9 and r3Sz raised the score to 6 reluctantly pointing out that this work lacks novel formulation and requires major re-organization/polishing. Whereas other reviewers iqhT and z46M were not satisfied with the response and retained their negative score.

---

### Decision · Program_Chairs · 2025-01-22

Reject